# Crystallochemical Design of Huntite-Family Compounds

**Galina M. Kuz'micheva [1], Irina A. Kaurova [1,\*] , Victor B. Rybakov [2] and Vadim V. Podbel'skiy [3]**

[1] MIREA-Russian Technological University, Vernadskogo pr. 78, Moscow 119454, Russia; kaurchik@yandex.ru

[2] Lomonosov Moscow State University named M.V. Lomonosov, Vorobyovy Gory, Moscow 119992, Russia; rybakov20021@yandex.ru

[3] National Research University ≪Higher School of Economics≫, Myasnitskaya str. 20, Moscow 101000, Russia; vpodbelskiy@hse.ru

\* Correspondence: kaurchik@yandex.ru; Tel.: +7-495-246-0555 (ext. 434)

**Abstract:** Huntite-family nominally-pure and activated/co-activated $LnM_3(BO_3)_4$ ($Ln$ = La–Lu, Y; $M$ = Al, Fe, Cr, Ga, Sc) compounds and their-based solid solutions are promising materials for lasers, nonlinear optics, spintronics, and photonics, which are characterized by multifunctional properties depending on a composition and crystal structure. The purpose of the work is to establish stability regions for the rare-earth orthoborates in crystallochemical coordinates (sizes of $Ln$ and $M$ ions) based on their real compositions and space symmetry depending on thermodynamic, kinetic, and crystallochemical factors. The use of diffraction structural techniques to study single crystals with a detailed analysis of diffraction patterns, refinement of crystallographic site occupancies (real composition), and determination of structure–composition correlations is the most efficient and effective option to achieve the purpose. This approach is applied and shown primarily for the rare-earth scandium borates having interesting structural features compared with the other orthoborates. Visualization of structures allowed to establish features of formation of phases with different compositions, to classify and systematize huntite-family compounds using crystallochemical concepts (structure and superstructure, ordering and disordering, isostructural and isotype compounds) and phenomena (isomorphism, morphotropism, polymorphism, polytypism). Particular attention is paid to methods and conditions for crystal growth, affecting a crystal real composition and symmetry. A critical analysis of literature data made it possible to formulate unsolved problems in materials science of rare-earth orthoborates, mainly scandium borates, which are distinguished by an ability to form internal and substitutional ($Ln$ and Sc atoms), unlimited and limited solid solutions depending on the geometric factor.

**Keywords:** huntite family; rare-earth scandium borate; optical material; crystal growth; rare-earth cations; X-ray diffraction (XRD); crystal structure; solid solution; order–disorder

## 1. Introduction

Modern scientific and applied materials science requires both an appearance of new materials with a desired combination of functional properties and an improvement and optimization of physical parameters and structural quality of the known materials, which have already proven themselves in practice, with further control of their properties using external (growth and post-growth treatment conditions) or internal (activation, isomorphic substitution) effects. Isomorphic substitutions and activation are different in the concentration of ion(s) introduced into a crystal matrix and effects produced. Isomorphic substitution is a powerful and flexible way to obtain a desired physical parameter of material by a targeted change in the composition of specific crystal structure. Activation is one of the necessary and relatively simple technological actions for the appearance, modification, and improvement of crystal properties by introduction (sometimes, over stoichiometry) of small amounts

of impurity ions into a crystal matrix that contributes to a self-organization and self-compensation of system's electroneutrality. It is possible that dopant ions introduced into a crystal structure in low concentrations over stoichiometry are distributed over crystallographic sites in a different way than those introduced in high concentrations in the case of formation of solid solutions. However, low content of dopant ions can lead to significant changes in both local and statistical structures, in particular, in crystal symmetry. In this paper, all the above-mentioned aspects are reviewed for the huntite family compounds, interesting and important materials from both applied and scientific points of view, with the main focus on single-crystal objects, a detailed study of which can obtainin reliable information about the internal structure of materials.

Complex orthoborates of rare-earth metals with the general chemical formula $LnM_3(BO_3)_4$, where $Ln^{3+}$ = La–Lu, Y and $M^{3+}$ = Al, Ga, Sc, Cr, Fe, belong to the huntite family (huntite $CaMg_3(CO_3)_4$, space group $R32$ [1]). Depending on the composition and external conditions, they can have both monoclinic (space groups $C2/c$, $Cc$, and $C2$) and trigonal (space groups $R32$, $P321$, and $P3_12$) symmetry with the presence (space group $C2/c$) or absence of center of symmetry.

The most widely known and studied representatives of the huntite family compounds are aluminum borates $LnM_3(BO_3)_4$ with the $M^{3+}$ = Al. The $LnAl_3(BO_3)_4$ crystals with the $Ln^{3+}$ = Y, Gd, Lu doped with the Nd, Dy, Er, Yb, Tm (for example, [2,3]) and co-doped with the Er/Tb, Er/Yb, Nd/Yb ions (for example, [4,5]), as well as single-crystal solid solutions in the $YAl_3(BO_3)_4$ – $NdAl_3(BO_3)_4$ system [6], are promising materials for self-frequency-doubled lasers. $YAl_3(BO_3)_4$ crystals doped with the $Er^{3+}$ or $Yb^{3+}$ ions are widely used in medicine and telecommunications as laser materials with a wavelength of 1.5–1.6 μm [4,5]. The $LnAl_3(BO_3)_4$ crystals with the $Ln^{3+}$ = Tb, Ho, Er, Tm exhibit a magnetoelectric effect ($HoAl_3(BO_3)_4$ is a leader among these compounds) (for example, [2,7]). The $LnAl_3(BO_3)_4$ compounds with the $Ln^{3+}$ = Gd, Eu, Tb, Ho, Pr, Sm are used as phosphors (for example, [8,9]): the $YAl_3(BO_3)_4$ crystals doped with the $Eu^{3+}$ and $Tb^{3+}$ ions is an environmentally friendly material for a white LED, having high intensity and luminescence power and low cost [10]; those doped with the $Tm^{3+}$ and $Dy^{3+}$ are able to be tuned from blue through white and ultimately to yellow emission colors and are attractive candidates for general illumination [11]; those doped with the $Sm^{3+}$ ions, as well as $GdAl_3(BO_3)_4$:$Sm^{3+}$, can be used as promising materials for orange-red lasers [8]. The simultaneous generation of three basal red–green–blue colors is obtained from a lone $GdAl_3(BO_3)_4$:$Nd^{3+}$ bi-functional laser and optical nonlinear crystal [12], which is also of interest for the development of high-quality and bright displays.

Rare-earth gallium borates $LnM_3(BO_3)_4$ with the $M^{3+}$ = Ga are poorly studied. The $LnGa_3(BO_3)_4$ crystals, in particular $LnGa_3(BO_3)_4$:$Tb^{3+}$ with the $Ln^{3+}$ = Y or Gd, are prominent luminescent materials with plasma-discharge conditions, converting a vacuum ultraviolet radiation into a visible light that can be used in high-performance plasma display panels and television devices [13]. In addition, gallium borates can be considered as promising materials not only for luminescent and laser applications, but also for use in spintronics: a large magnetoelectric effect was found in the $HoGa_3(BO_3)_4$ [14].

Rare-earth borates $LnM_3(BO_3)_4$ with the magnetic ions $M^{3+}$ = Fe or Cr, which are characterized by the presence of two interacting magnetic subsystems (3$d$- and 4$f$- ions) in the crystal structure, are even less studied. A long-range antiferromagnetic spin order of the $Cr^{3+}$ subsystem is observed in the $NdCr_3(BO_3)_4$ single crystals at $T_N$ = 8 K [15]. A phase transition from paramagnetic to antiferromagnetic state was found in the $EuCr_3(BO_3)_4$ crystals at $T_N \approx 9$ K [16]. In addition, a magnetic phase transition is also observed in the $SmCr_3(BO_3)_4$ at $T_c = 5 \pm 1$ K [17]. In a number of rare-earth ferroborates, a significant magnetoelectric effect was found [18,19], which makes it possible to attribute them to a new class of multiferroics [18]. Maximum magnetoelectric and magnetodielectric effects were recorded for the $NdFe_3(BO_3)_4$ and $SmFe_3(BO_3)_4$ crystals [19,20]. Such materials can be used as magnetoelectric sensors, memory elements, magnetic switches, spintronics devices, high-speed radiation-resistant MRAM memory, etc.

Rare-earth scandium borates $LnM_3(BO_3)_4$ with the $M^{3+}$ = Sc demonstrate an anomalously low luminescence concentration quenching, which is caused by the large distance between the nearest

*Ln* ions (~6 Å), and, hence, these compounds are promising high-efficient optical media that can be used in photonics, in particular, to create a diode-pumped compact lasers covering various optical spectral regions [21,22]. Currently, the best materials for medium-power solid-state miniature lasers are undoped and $Cr^{3+}$-doped $NdSc_3(BO_3)_4$ [22,23]. They are characterized by a high efficiency of laser transitions, on the one hand, and detuning of almost all cross-relaxation transitions, on the other. In addition, the $NdSc_3(BO_3)_4$ crystals have a high non-linear dielectric susceptibility. Hence, in these crystals, it is possible to convert an infrared radiation of the neodymium laser into a visible one along a certain crystal orientation relative to the propagation of laser radiation. A miniature laser emitted in a green spectral region can be created based on the $NdSc_3(BO_3)_4$.

As can be seen from the short literature review on properties and possible areas of application of the huntite-family borates, these materials are characterized by a combination of different functional properties in one compound. In turn, physical and chemical properties are due to a real composition and crystal structure, influenced by initial composition (the type of rare-earth metal and the nature of *M* metal), synthesis conditions, and external effects.

Determination of structure of huntite-family rare-earth borates becomes very important, since a possible structural transition from one space group to another can be accompanied by a loss (or acquisition) of the center of symmetry and results in an acquisition (or loss) of nonlinear optical and magnetoelectric properties. Sardar et al. [24] found that the introduction of 5 at % $Nd^{3+}$ ions ($2.3 \times 10^{20}$ cm$^{-3}$; activation) into the $LaSc_3(BO_3)_4$ crystal leads to a change in symmetry from the space group *C*2/*c* to *C*2, i.e., to a transition from a centrosymmetric to a non-centrosymmetric structure. In addition, it was shown that the introduction of $Nd^{3+}$ ions in a concentration more than or equal to 50 at % to the $LaSc_3(BO_3)_4$ crystal (solid solution) results in the space group transition from *C*2/*c* to *R*32 [25]. A comprehensive study of solid solutions in the system $NdCr_3(BO_3)_4$ (sp. gr. *C*2/*c*)–$GdCr_3(BO_3)_4$ (sp. gr. *R*32) using spectroscopic methods showed that the $Nd_xGd_{1-x}Cr_3(BO_3)_4$ borates with x < 0.2 have essentially trigonal non-centrosymmetric structure (sp. gr. *R*32), and already in the case of 20% Nd concentration in the crystals, an additional large content of the monoclinic phase (sp. gr. *C*2/*c*) is observed [26].

Knowledge of a precise real composition of crystals is no less important. Bulk crystals, usually obtained by the most technological melt methods, can have a homogeneous composition only for compounds with the congruent melting (**CM**). However, an activation of crystal or a synthesis of mixed crystals (solid solutions) leads to a deviation of a real composition from the CM one. In addition, a real composition of grown crystal, taking into account compositions of all crystallographic sites, usually differ from that of initial charge (melt). However, in the overwhelming majority of cases, observed physical properties of materials are 'attributed' to initial charge composition that leads to incorrect conclusions about correlations in the fundamental triad 'composition-structure-properties'.

Thus, to improve properties of huntite-family single crystals and expand the scope of their application as well as to synthesize and create materials with a required combination of operating parameters, it is necessary to know a precise real crystal composition, structural effects, and crystallochemical limits of existence (stability limits) of a compound or solid solution with a specific symmetry. This is the motivating force for the research, the results of which are reported on here.

Currently, various approaches are developed to avoid time-consuming searches for materials with a specific crystal structure and desired physical parameters and to optimize and simplify ways of their obtaining. One of the most effective techniques is a type of crystal engineering which is shown here for the huntite-family compounds. In this approach, the 'composition-structure-property' correlations for a specific class of materials as well as possible structural types with chemical element sets formed crystal structure, coordination environment of atoms in the structure, nature of chemical bond between different atomic groups, etc. are analyzed. When forecasting new structures or describing the known ones, it is necessary to take into account crystallochemical characteristics of atoms or ions in the structure, namely, a radius, an electronegativity, a formal charge, and hence a coordination number and bonds between the components. An approach described is one of the main driving forces in the applied

crystallochemistry—a section of materials science, which combines knowledges from fundamental crystallochemistry and specialists in other scientific fields dealing with materials. One of the main tasks of applied crystallochemistry is an investigation of functional correlations of the form of $P = P(X)$, where $X$ is a material and $P$ is a property. In this case, properties are considered depending on the crystallochemical individuality of the components in a number of related compounds. A computer design of such structures involves a generation of polyhedra, a search for possible packages with the following determination of all required structural parameters.

## 2. Materials and Methods

In this review, single crystals are predominantly described and a preference is given to X-ray diffraction research techniques—a powerful tool to determine fundamental characteristics of an object, in particular, a real composition and structure, with a high degree of accuracy and reliability and to reasonably relate functional properties to a composition and structure depending on the prehistory of crystals (composition of initial charge, growth and post-growth treatment conditions). A crystallochemical approach based on the knowledge of basic laws of structure formation and correlations between real composition, structure, growth conditions, and physical properties of materials, makes it possible to optimize a search for new promising compounds and to improve functional properties of the known ones. The use of information technologies applied crystallochemical models for a subsequent creation of mathematical models and specialized programs on their basis accelerates a transition from theory to practice, and further to a technology to grow material with controlled properties.

The $LnM_3(BO_3)_4$ crystals, where $Ln^{3+}$ = La–Lu, Y, $M^{3+}$ = Al, Fe, Ga, Cr, melt incongruently and are usually obtained by the flux method from the high-temperature solutions both via spontaneous crystallization and using a seed, which is described in detail in [21,27–29]. The use of the flux method, as the most suitable for growing such crystals, has a significant drawback—an incorporation of flux components into the growing crystal. The $K_2Mo_3O_{10}$–$B_2O_3$ mixed flux is usually applied to grow huntite-family borates [21,27,30], however, the Mo ions easily incorporate into a crystal, which leads to an appearance of a near ultraviolet absorption band that inhibits the use of these nonlinear optical crystals at short wavelength [21,27,31]. In addition, crystals have low growth rate and small sizes (1–20 mm), which leads to an impossibility of obtaining bulk samples of good optical quality suitable for further use. In [21,32,33] only, the relatively large $LnM_3(BO_3)_4$ crystals, in particular, $YAl_3(BO_3)_4$ and $GdAl_3(BO_3)_4$, up to 45 mm in size have been synthesized by the top-seeded solution growth (**TSSG**) method.

In addition, it should be noted that the polycrystalline gallium borates with the $Ln$ = La, Nd, Sm, Gd, Ho, Y, Er, and Yb were sintered in Pt crucible in air atmosphere at $T$ = 575–1050 °C from pellets made from $Ln_2O_3$, $Ga_2O_3$, and $B_2O_3$ oxide powders [34]. As a result, the dominant metaborate $Ln(BO_2)_3$ for the $Ln$ = La and Nd, huntite $LnGa_3(BO_3)_4$ for the $Ln$ = Sm, Gd, Ho, Y and Er, the new dolomite $YbGa(BO_3)_2$, the intermediate $LnBO_3$ and $GaBO_3$ phases were identified by X-ray powder diffraction measurements.

A significant advantage of the $LnSc_3(BO_3)_4$ scandium borates is a possibility of synthesis of large-sized single crystals using the Czochralski technique ($Ln$ = La, Ce, Pr, Nd; in particular, [35]). Wherein, the degree of congruence of $LnSc_3(BO_3)_4$ decreases from La to Nd [35], and compounds with the $Ln$ = Y and Gd melt incongruently [36,37]. Durmanov et al. [33] developed the modified heating assembly served to effectively control both the process of possible condensation of $B_2O_3$ vapors on the surface of the growing crystal and thermal gradients over the crucible and in the melt. It guaranteed synthesis of high-quality optical $LaSc_3(BO_3)_4$ single crystals (both pure and doped with the Nd, Yb, Er/Yb, Er/Yb/Cr, Pr), $CeSc_3(BO_3)_4$, $PrSc_3(BO_3)_4$ (both pure and doped with the Nd), $NdSc_3(BO_3)_4$, as well as solid solutions, in particular, those with the general chemical compositions $(Ce,Nd)Sc_3(BO_3)_4$, $(Ce,Gd)Sc_3(BO_3)_4$, $(Nd,Gd)Sc_3(BO_3)_4$, $(Ce,Nd,Gd)Sc_3(BO_3)_4$, $(Ce,Y)Sc_3(BO_3)_4$, $Ce(Lu,Sc)_3(BO_3)_4$, $(Ce,Nd)(Lu,Sc)_3(BO_3)_4$.

Single crystals with the general composition $LnSc_3(BO_3)_4$ ($Ln$ = La, Ce, Pr, Nd, Tb), in particular, with the initial charge compositions $LaSc_3(BO_3)_4$, $CeSc_3(BO_3)_4$, $Pr_{1.1}Sc_{2.9}(BO_3)_4$, $Pr_{1.25}Sc_{2.75}(BO_3)_4$, $NdSc_3(BO_3)_4$, $Nd_{1.25}Sc_{2.75}(BO_3)_4$, $TbSc_3(BO_3)_4$ and single-crystal solid solutions with the initial compositions $(La,Nd)Sc_3(BO_3)_4$, $(Ce,Nd)_{1+x}Sc_{3-x}(BO_3)_4$, $(La,Pr)Sc_3(BO_3)_4$, $La(Sc,Yb)Sc_3(BO_3)_4$, $La(Sc,Er)Sc_3(BO_3)_4$, $La(Sc,Er,Yb)Sc_3(BO_3)_4$, $(Ce,Gd)Sc_3(BO_3)_4$, $(Ce,Gd)_{1+x}Sc_{3-x}(BO_3)_4$, $(Nd,Gd)_{1+x}Sc_{3-x}(BO_3)_4$, $(Ce,Nd,Gd)Sc_3(BO_3)_4$, $(Ce,Y)_{1+x}Sc_{3-x}(BO_3)_4$, $(Ce,Lu)_{1+x}Sc_{3-x}(BO_3)_4$, $(Ce,Nd,Gd)_{1+x}Sc_{3-x}(BO_3)_4$ $(Ce,Nd,Lu)_{1+x}Sc_{3-x}(BO_3)_4$, described in this work, having averaged diameter of 15–25 mm and length of 30–150 mm were grown by the Czochralski technique in Ir crucibles at pulling rate 1–3 mm/h and seed rotation 8–12 rpm. An Ir rod of 2 mm in diameter was initially used as a seed. Oriented single-crystal seeds were cut from the crystals grown on the Ir rod. The seed was oriented so that its optical axis coincided with the pulling axis (within a few degrees). The methodology for Czochralski growth technique used to synthesis $LnSc_3(BO_3)_4$ crystals ($Ln$ = La, Ce, Pr, Nd, Tb) as well as single-crystal solid solutions is given in [38,39].

According to the literature data, a crystallization of $LnM_3(BO_3)_4$ compounds ($Ln^{3+}$ = La–Lu, Y; $M^{3+}$ = Al, Fe, Ga, Cr) in a specific space group was found using different techniques, namely, diffraction methods (**D**), infrared spectroscopy (**IR**), absorption spectroscopy (**AS**), transmission spectroscopy (**TS**), Raman spectroscopy (**R**); in addition, temperatures of structural phase transitions between forms with different space groups were determined using specific heat measurements (**SH**) and differential thermal analysis (**DTA**) [1,15,16,26,27,34–37,40–89] (Table 1). In several works [36,49], the structures of the compounds were declared solely on the basis of a comparison of the experimental diffraction patterns with those given in the structural databases without any detailed structural analysis. In the overwhelming number of cases, the crystal structures (first of all, coordinates of atoms) of the samples were refined by the full-profile Rietveld method on polycrystalline samples obtained by the solid-state reaction ($LnFe_3(BO_3)_4$ with the $Ln$ = La, Ce, Pr, Nd, Sm, Eu, Gd, Tb, Dy, (Y), Ho; $LnGa_3(BO_3)_4$ with the $Ln$ = Sm, Gd, Y, Ho, Er) [34,53] or on single crystals synthesized by the flux method and ground to a powder ($LnAl_3(BO_3)_4$ with the $Ln$ = Nd, Sm, Eu, Gd, Tb, Dy, (Y), Ho, Er, Yb; $LnFe_3(BO_3)_4$ with the $Ln$ = Ce, Pr, Nd, Sm, Eu, Gd, Tb, Dy, (Y), Ho; $LnGa_3(BO_3)_4$ with the $Ln$ = Gd) [1,54,56,59,61,66,68]. Assignment of a series of rare-earth orthoborates ($LnAl_3(BO_3)_4$ with the $Ln$ = Nd, Eu, Gd, Tb, Dy, (Y), Ho, Er, Tm, Yb; $LnFe_3(BO_3)_4$ with the $Ln$ = Eu, Gd, Y; $LnCr_3(BO_3)_4$ with the $Ln$ = Sm, Eu, Gd, Tb, Dy; $LnGa_3(BO_3)_4$ with the $Ln$ = Nd, Eu, Gd, (Y), Ho) to the space group $R32$ is performed on powdered single crystals obtained by the flux method using IR spectroscopy coupled with the group-theoretical analysis [26,42,45,51,70,71,73], temperature-dependent high-resolution optical absorption Fourier spectroscopy and Raman spectroscopy ($LnFe_3(BO_3)_4$ with the $Ln$ = Pr, Nd) [57,58,60].

Crystal structures of single crystals were refined within the framework of the huntite structure (space group $R32$) for the $LnAl_3(BO_3)_4$ ($Ln$ = Nd, Sm, Eu, Gd, (Y), Tm, Yb), $LnFe_3(BO_3)_4$ ($Ln$ = La, Nd, Eu, Gd, Er), $LnGa_3(BO_3)_4$ ($Ln$ = Nd, Eu, Ho), $LnSc_3(BO_3)_4$ ($Ln$ = La, Ce, Pr, Nd, Sm, Eu) by the X-ray diffraction (**XRD**) analysis (Table 1). It should be noted that in some literature sources (in particular, in [27]) it was not indicated on which samples, single-crystal or polycrystalline, a structural analysis has been performed; in [15,36,37,80], any methodology of structural studies is absent, the method of investigation, X-ray diffraction, being indicated only. Neutron diffraction study performed on powdered samples with the initial compositions $YAl_3(BO_3)_4$, $Y_{0.88}Er_{0.12}Al_3(BO_3)_4$, $Y_{0.5}Er_{0.5}Al_3(BO_3)_4$, $Y_{0.5}Yb_{0.5}Al_3(BO_3)_4$, and $Y_{0.84}Er_{0.01}Yb_{0.15}Al_3(BO_3)_4$, grown by the topseeded high temperature solution method, crystallized in the space group $R32$ [90]. Due to the fact that the neutron scattering amplitudes both for Er (b = 7.79 fm) and for Yb (b = 12.433 fm) are much greater than that for Al (b = 3.449 fm), according to the Rietveld calculations, it can be stated that the $Er^{3+}$ and $Yb^{3+}$ ions occupy the $Y^{3+}$ sites. For the above-mentioned compounds, positional parameters have been refined only [90].

**Table 1.** Space groups for the compounds with the general composition $LnM_3(BO_3)_4$ with the $M^{3+}$ = Al, Fe, Cr, Ga, Sc (according to the literature data).

| Ln | Space groups for the $LnM_3(BO_3)_4$ [1] | | | | |
|---|---|---|---|---|---|
| | *M* = Al | *M* = Fe | *M* = Cr [2] | *M* = Ga | *M* = Sc |
| La | **Orthorombic symmetry:** D/P [40] | *R32*: D/? [27], D/P [53,54], D/S [55] | *C2/c*: IR/S (70:30, 1040–1050 °C) [70], IR/S (1.5:1; 2.3:1) [71] | | *R32*: D/S [74,75] *C2/c*: D/S (or *C2*) [76], D/S [77], D/P [78] *Cc*: D/S [79] |
| Ce | | *R32*: D/P [53] | | | *R32*: ? [80], D/S [81] *C2/c*: D/S [35,77,82–87] *Refined composition:* $\overline{CeSc_3(BO_3)_4}$ [82] |
| Pr | *R32*: D/? [27] *C2/c*: D/? [27], D/S [41], IR/P [42] *C2*: D/? [27] | *R32*: D/? [27], D/P [53], D/P (1.5, 300 K) [56], AS/S [57,58] | *C2/c*: IR/S (50:50, 900–950 °C; 70:30, 1040–1050 °C) [70], IR/S (1:1; 1.5:1; 2.3:1) [71] | | *R32*: ? [80], D/S [88] *C2/c* (or *C2*): D/S [35,83] *P321*: D/S (40% reflections *R32*) [82] *Refined compositions:* $(Pr_{0.919}Sc_{0.081(4)})Sc_3(BO_3)_4$ [82], $(Pr_{0.924}Sc_{0.076(4)})Sc_3(BO_3)_4$ [82] |
| Nd | *R32*: D/P [1], D/? [27], D/S [43,44], IR/P [42,45] *C2/c*: D/S [46,47], IR/P [42,45], D/P [48], ? [49] *C2*: D/P [48], ? [49] | *R32*: D/? [27], D/P [53,54,59], D/S [55], R/S [60], AS/S [57,58] *C2/c*: IR/P [45] | *R32*: D/? [27], ? [15] *C2/c*: IR/S (50:50, 900–950 °C; 70:30, 1040–1050 °C) [70], IR/S (1:1; 2.3:1) [71], IR/S [26,45] | *R32*: D/S [47], D/? [27], IR/P [45] | *R32*: ? [80], D/S [77], D/S [88] *P321* (or *P3*): D/S [86,87,89] *P321*: D/S (40% reflections *R32*) [82] *Refined compositions:* $NdSc_3(BO_3)_4$ [82,89], $(Nd_{0.910}Sc_{0.090(20)})Sc_3(BO_3)_4$ [82] |
| Sm | *R32*: D/P [1], D/? [27], D/S [50] *C2/c*: D/? [27], IR/P [42] | *R32*: D/? [27], D/P [53,54,61] | *R32*: D/? [27], IR/S (1:1) [71], IR/S (50:50, 900–950 °C) [70] *C2/c*: IR/S (70:30, 1040–1050 °C) [70], IR/S [72] *R32 + C2/c* fragments: IR/S (1.5:1) [71] *C2/c + R32*: IR/S (2.3:1) [71] | *R32*: D/? [27], D/P [34] | *R32*: ? [80], D/S [88] |
| Eu | *R32*: D/P [1], D/? [27], D/S [50], IR/P [42,51] *C2/c*: D/? [27] *C2*: D/S [41] | *R32*: D/? [27], D/S [62], D/P [53,54], IR/P [51] *R32* (HT) and *P3₁21* (LT): SH/S, $T_s$ = 88 K [53]; AS/S, $T_s$ = 58 K [57,58]; TS/S, $T_s$ = 84 K [62] | *R32*: D/? [27], D/P [16] IR/S (50:50, 900–950 °C; 70:30, 1040–1050 °C) [70], IR/S (1:1; 2.3:1) [71], IR/S [51] | *R32*: D/? [27], D/S [73], IR/P [51,73] | *R32*: D/S [88] |
| Gd | *R32*: D/P [1], D/? [27], D/S [50], D/P [52], IR/P [42,45] *C2*: D/S [46,47] | *R32*: D/? [27], D/P [34,54], D/S (297 K) [63], IR/P [45], *P3₁21*: D/S (90 K) [63] *R32* (HT) and *P3₁21* (LT): SH/S, $T_s$ = 174 K [53]; R/S, $T_s$ = 155 K [60]; AS/S, $T_s$ = 133–156 K [57,58,64]; IR/S, $T_s$ = 143 K [65] | *R32*: D/P [1], D/? [27], IR/S (50:50, 900–950 °C; 70:30, 1040–1050 °C) [70], IR/S (1:1; 1.5:1; 2.3:1) [71], IR/S [26,45] | *R32*: D/? [27], D/P [34,54], IR/P [45] | *R32*: ? [36,37] |

**Table 1.** *Cont.*

| Ln | Space groups for the $LnM_3(BO_3)_4$ [1] | | | | |
|---|---|---|---|---|---|
| | *M* = Al | *M* = Fe | *M* = Cr [2] | *M* = Ga | *M* = Sc |
| Tb | **R32:** D/P [1], D/? [27], IR/P [42] **C2/c:** D/S [41] | **R32:** D/? [27], D/P [53,54], D/P (200 K, 300 K) [66] **P3₁21:** D/P (2, 30, 40, 100 K) [66] **R32 (HT) and P3₁21 (LT)** SH/S, T$_s$ = 241 K [53]; R/S, T$_s$ = 198 K [60]; AS/S, T$_s$ = 198 K [57,58], T$_s$ = 192 K [66]; IR/S, T$_s$ = 200 K [65] | **R32:** D/? [27], IR/S (50:50, 900–950 °C) [70] **R32 + C2/c fragments:** IR/S (1:1) [71] **C2/c + R32 fragments:** IR/S (1.5:1) [71] **C2/c:** IR/S (70:30, 1040–1050 °C) [70], IR/S (2.3:1) [71] | **R32:** D/? [27] | **R$\overline{3}$ or R3-Calcite-type structure:** D/S [35,83] *Refined composition:* $(\overline{Tb_{0.25}Sc_{0.75}})BO_3$ [83] |
| Dy | **R32:** D/P [1], D/? [27], IR/P [42] | **R32:** D/? [27], D/P [53,54], **P3₁21:** D/P (1.5, 50, 300 K) [67] **R32 (HT) and P3₁21 (LT):** SH/S, T$_s$ = 340 K [53] | **R32:** IR/S (50:50, 900–950 °C) [70], IR/S (1.5:1) [71] **R32 + C2/c fragments:** IR/S (1:1) [71] **C2/c:** IR/S (70:30, 1040–1050 °C) [70], IR/S (2.3:1) [71] | **R32:** D/? [27] | |
| (Y) | **R32:** D/P [1], D/S [46,47], D/? [27], IR/P [42,45] | **R32:** D/? [27], D/P [53,54], IR/P [45], D/P (520 K) [68] **P3₁21:** D/P (2, 50, 295 K) [68] **R32 (HT) and P3₁21 (LT):** DTA/S, T$_s$ = 445 K [53]; R/S, T$_s$ = 350 K [60]; AS/S, T$_s$ = 350 K [57,58] | | **R32:** D/? [27], D/P [34], IR/P [45] | **R32:** ? [36,37] |
| Ho | **R32:** D/P [1], D/? [27], IR/P [42] **C2/c:** D/S [41] | **R32:** D/? [27], D/P [53,54], D/P (520 K) [68] **P3₁21:** D/P (2, 50, 295 K) [68] **R32 (HT) and P3₁21 (LT):** DTA/S, T$_s$ = 427 K [53] | **R32:** D/? [27] **R32 + C2/c fragments:** IR/S (1.5:1) [71], **C2/c:** IR/S (70:30, 1040–1050 °C) [70], IR/S (2.3:1) [71] | **R32:** D/? [27], D/S [73], D/P [34], IR/P [73] | |
| Er | **R32:** D/P [1], D/? [27], IR/P [42] | **R32:** D/? [27], D/S [69] **P3₁21:** D/P (1.5, 300 K) [56] **R32 (HT) and P3₁21 (LT):** R/S, T$_s$ = 340 K [60]; AS/S, T$_s$ = 340 K [57,58] | **R32 + C2/c fragments:** IR/S (1:1) [71] | **R32:** D/? [27], D/P [34] | |
| Tm | **R32:** D/? [27], D/S [41], IR/P [42] | **R32:** D/? [27] | | | |
| Yb | **R32:** D/P [1], D/? [27], D/S [41], IR/P [42] | **R32:** D/? [27] | **R32:** D/? [27] | **R32:** D/? [27] **R3-Dolomite type structure:** D/P [34] | |
| Lu | **R32:** D/? [27] | | | | |

[1] P—powder sample, S—single crystal sample, D—diffraction techniques, IR—infrared spectroscopy, AS—absorption spectroscopy, TS—transmission spectroscopy, R—Raman spectroscopy, SH—specific heat measurements, DTA—differential thermal analysis, HT—high-temperature phase, LT—low-temperature phase, T$_s$—phase transition temperature; A question mark (?) indicates a lack of data on the type of material under investigation or/and applied technique in the literature source. [2] The borate:solvent ratios in the batch and the temperature ranges are given in parentheses.

Resuts of XRD study of the Czochralski-grown single crystals and solid solutions ("Enraf-Nonius" CAD-4 single-crystal diffractometer; room temperature; $AgK_\alpha$, $MoK_\alpha$ or $CuK_\alpha$; size, $\sim 0.1 \times 0.1 \times 0.1$ mm$^3$) with the initial compositions $LaSc_3(BO_3)_4$, $CeSc_3(BO_3)_4$, $Pr_{1.1}Sc_{2.9}(BO_3)_4$, $Pr_{1.25}Sc_{2.75}(BO_3)_4$, $NdSc_3(BO_3)_4$, $Nd_{1.25}Sc_{2.75}(BO_3)_4$, $TbSc_3(BO_3)_4$ and $(La,Nd)Sc_3(BO_3)_4$, $(Ce,Nd)_{1+x}Sc_{3-x}(BO_3)_4$, $(La,Pr)Sc_3(BO_3)_4$, $La(Sc,Yb)Sc_3(BO_3)_4$, $La(Sc,Er)Sc_3(BO_3)_4$, $La(Sc,Er,Yb)Sc_3(BO_3)_4$, $(Ce,Gd)Sc_3(BO_3)_4$, $(Ce,Gd)_{1+x}Sc_{3-x}(BO_3)_4$, $(Nd,Gd)_{1+x}Sc_{3-x}(BO_3)_4$, $(Ce,Nd,Gd)Sc_3(BO_3)_4$, $(Ce,Y)_{1+x}Sc_{3-x}(BO_3)_4$, $(Ce,Lu)_{1+x}Sc_{3-x}(BO_3)_4$, $(Ce,Nd,Gd)_{1+x}Sc_{3-x}(BO_3)_4$ $(Ce,Nd,Lu)_{1+x}Sc_{3-x}(BO_3)_4$, performed by our scientific group, are given in [35,76,82,83,85,89,91] and [35,84,86,87,91], respectively, and in the present work. To reduce an error associated with an absorption, the XRD data were collected over the entire Ewald sphere. The unit cell parameters are determined by an auto-indexing of the most intense 25 reflections. Furthermore, a detailed analysis of diffraction reflections, including low-intensity ones, was carried out to find a possible superstructure either with the multiple-increased unit cell parameters and another symmetry or with another symmetry only. In the case of a small number of reflections that do not obey the extinction rules of the chosen space group, these reflections were not taken into account when refining a crystal structure. In case of a large number of 'forbidden' reflections, a crystal structure was solved by direct methods or/and the Paterson method taking into account all diffraction reflections. The preliminary XRD data processing was carried out using the WinGX pack [92] with a correction for absorption. The atomic coordinates, anisotropic displacement parameters of all atoms, and occupancies of all the sites (except for the B and O ones) were refined using the SHELXL2013 software package [93], taking into account the atomic scattering curves for neutral atoms. The structural parameters were refined in several steps: initially, the coordinates of 'heavy' atoms (*Ln* and Sc), and then those of 'light' atoms (O and then B) were refined together with the atom displacement parameters in isotropic and then anisotropic approximations; finally, the occupancies of the *Ln* and Sc sites were refined step by step.

In the review, for the visualization and comparison of crystal structures (ball-and-stick and polyhedral models) and individual polyhedra as well as for the calculation of structural parameters (all interatomic distances, bond angles, etc.) of the huntite-family compounds and solid solutions with different symmetry, the improved and augmented computer program for the investigation of the dynamics of changes in structural parameters of compounds with different symmetry has been applied [94]. Theoretical diffraction patterns have been created using the DIAMOND [95] and specialized software developed for diffraction pattern indexing taking into account a selection of background level for rhombohedral and hexagonal cells of the huntite structure and a refinement of unit cell parameters using different sets of diffraction reflections: CPU, Intel core i3; RAM, at least 4 GB; Code, C#; OS: Windows 7 with the installed Microsoft.NET Framework 4.0 or higher; Size: 32 768 b.

## 3. Results

### 3.1. $LnM_3(BO_3)_4$ (M = Al, Fe, Cr, Ga, Sc) Compounds

All the known literature data on symmetry of the huntite-family compounds, determined mainly by diffraction and spectroscopic methods on both polycrystalline and single-crystal samples, are systematized and given in Table 1 and Figure 1. It can be noted that almost all compounds having the general composition $LnM_3(BO_3)_4$ with the $M^{3+}$ = Al, Fe, Cr, Ga, Sc have a modification with the huntite structure (space group $R32$).

The situation looks different if only the results of X-ray study of single crystals are taken into account (Figure 2): the compounds with the general composition $LnM_3(BO_3)_4$ with the $M^{3+}$ = Al и Sc are most fully represented; a modification with the huntite structure (space group $R32$) prevails.

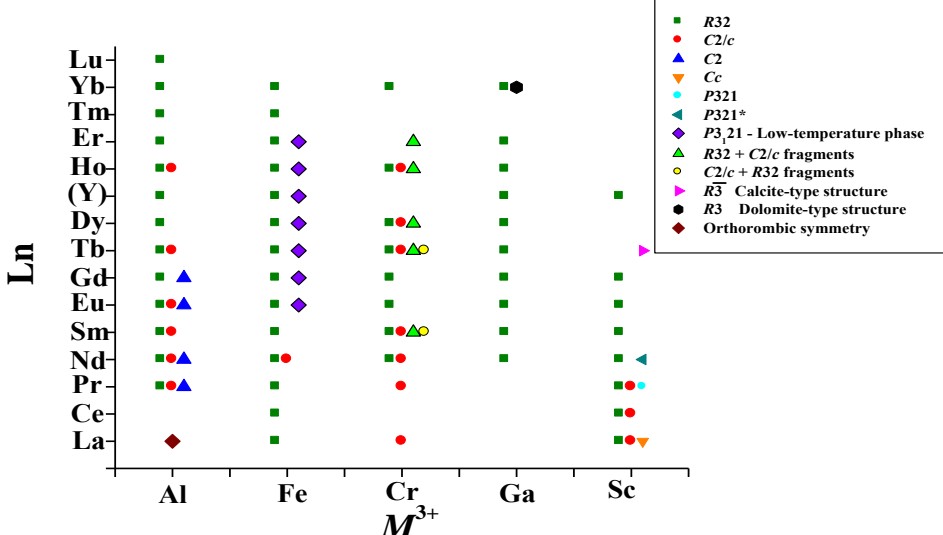

**Figure 1.** Space groups for the compounds with the general composition $LnM_3(BO_3)_4$ with the $M^{3+}$ = Al, Fe, Cr, Ga, Sc (according to the literature data given in Table 1).

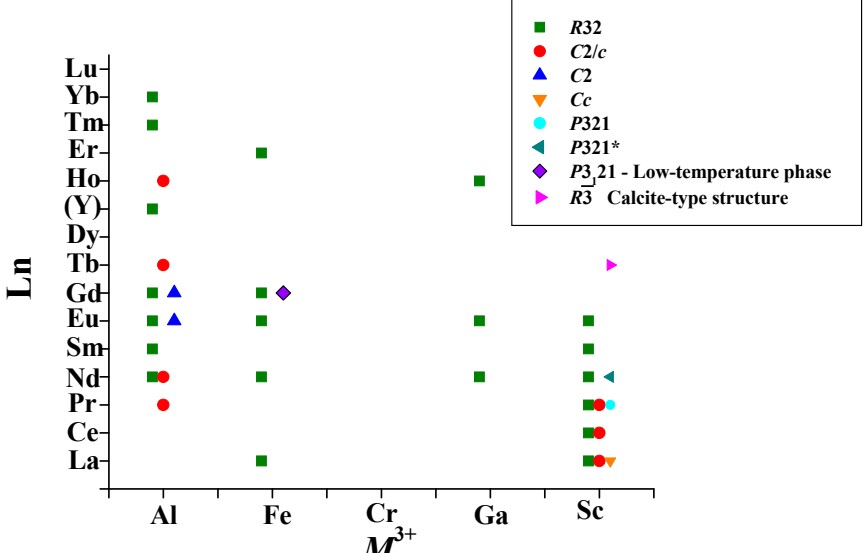

**Figure 2.** Space groups for the single crystals with the general composition $LnM_3(BO_3)_4$ with the $M^{3+}$ = Al, Fe, Ga, Sc (according to the literature data on X-ray diffraction experiments given in Table 1).

In the crystal structure of the huntite $CaMg_3(CO_3)_4$ (space group $R32$, $a$ = 9.5027(6), $c$ = 7.8212(6) Å, $Z$ = 3) (Figure 3), the $Ca^{2+}$ ion ($r_{Ca}^{VI}$ = 1.00 Å according to the Shannon system [96]) is located in the center of a distorted prism with the CN Ca = 6 (CN, coordination number). The upper triangular face of prism is rotated with respect to the lower one by an angle φ = 9.596° (φ = 0° in the regular trigonal prism), the Ca–O interatomic distances being the same. The $Mg^{2+}$ ion ($r_{Mg}^{VI}$ = 0.72 Å) is located in the center of a distorted octahedron with three different Mg–O interatomic distances (CN Mg = 2 + 2 + 2). Crystallochemically-different B1 and B2 ions occupy the centers of isosceles (CN B1 = 1 + 2) and equilateral (CN B2 = 3) triangles, respectively. The $MgO_6$ octahedra are joined by the edges and form twisted chains extended parallel to the $c$ axis (the $3_1$ axis). The B2 atoms are located at the two-fold axes in triangles between the chains from the $MgO_6$ octahedra, forming a 'spiral staircase' around the $3_2$ axes. Different chains are connected by the $CaO_6$ trigonal prisms and $BO_3$ triangles, where each individual $CaO_6$ and $BO_3$ group connects three chains [55,65]. Figure 3 shows the XY and XZ projections of huntite-type unit cell of $LnM_3(BO_3)_4$ (space group $R32$) as a

ball-and-stick model (Figure 3a,b) and polyhedra (Figure 3c,d) as well as fragments of the structure, including the main coordination polyhedra (Figure 3e,f), and individual coordination polyhedra for all crystallochemically-different atoms in the crystal structure (Figure 3g–i).

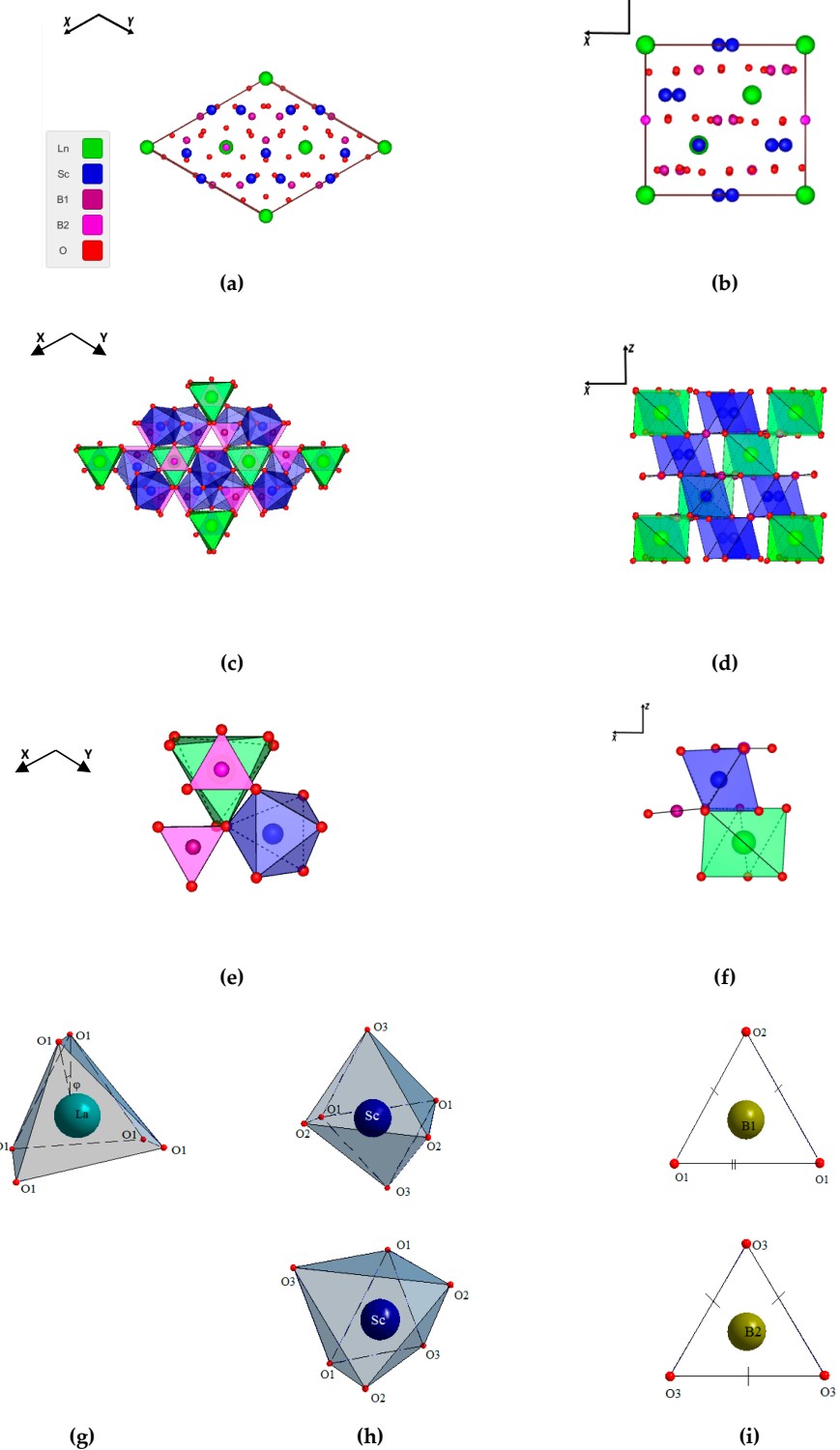

**Figure 3.** The unit cell of the $LnM_3(BO_3)_4$ structure (space group $R32$) projected onto the (**a**) XY and (**b**) XZ planes; Combination of the coordination polyhedra projected onto the (**c**) XY and (**d**) XZ planes; Combination of selected coordination polyhedra projected onto the (**e**) XY and (**f**) XZ planes; Coordination polyhedra for the (**g**) $Ln$, (**h**) $M$, (**i**) B1 and B2.

A topological correspondence of the $Ca^{2+}Mg^{2+}_3(CO_3)^{4-}_4$ and $Ln^{3+}M^{3+}_3(BO_3)^{3-}_4$ formulas, a similar coordination environment of the $C^{4+}$ and $B^{3+}$ ions, a possibility of the compensation of system electroneutrality, and a presence of the $Ca^{2+}$ and $Ln^{3+}$, $Mg^{2+}$ and $Sc^{3+}$, $Mg^{2+}$ and $M^{3+}$ = Al, Ga, Fe ions in the Goldschmidt–Fersman diagonal series should lead to the isostructurality of the $Ca^{2+}Mg^{2+}_3(CO_3)^{4-}_4$ and $Ln^{3+}M^{3+}_3(BO_3)^{3-}_4$ compounds (more precisely, these compounds are isotype, as evidenced by the lack of similarity in structures, typical of isostructural compounds [91]) despite the fundamental difference in the crystallochemical properties (dimensions, electronegativity values, formal charges).

It should be noted that the $Cr^{3+}$ ions are surrounded by six O atoms, forming the regular $CrO_6$ octahedra (CN Cr = 6) in the crystal structures of oxides due to the electronic structure (non-binding configuration is symmetric to the octahedral field of ligands - $d_\varepsilon^3$), unlike distorted $MO_6$ octahedra in the huntite structure (another example confirmed that the $Ca^{2+}Mg^{2+}_3(CO_3)^{4-}_4$ and $Ln^{3+}Cr^{3+}_3(BO_3)^{3-}_4$ compounds are isotype). Hence, in the structure of the activated $La^{3+}Sc^{3+}_3(BO_3)^{3-}_4$:$Cr^{3+}$ crystal, the symmetry of the $ScO_6$ polyhedra, which include $Cr^{3+}$ ions, as well as that of the whole crystal changes. This is confirmed by the XRD analysis of the $La^{3+}Sc^{3+}_3(BO_3)^{3-}_4$ and $La^{3+}Sc^{3+}_3(BO_3)^{3-}_4$:Cr single crystals, grown by the Czochralski method, with the monoclinic (space group $C2/c$) and triclinic (space group $P1$ or $P\bar{1}$) symmetry, respectively [76]. In the latter case, quite a lot of reflections with the $I < 3\sigma(I)$ are not indexed in the monoclinic syngony, and taking into account these reflections, the refined unit cell parameters of the $LaSc_3(BO_3)_4$:Cr were found to be $a = 7.7356(4)$, $b = 9.8533(8)$, $c = 12.0606(8)$ Å, $\alpha = 89.981(6)$, $\beta = 105.437(5)$, $\gamma = 90.045(6)°$, in contrast to those of the $LaSc_3(BO_3)_4$, $a = 7.727(1)$, $b = 9.840(1)$, $c = 12.046(3)$ Å, $\beta = 105.42(2)°$.

In the $CaCO_3$-$MgCO_3$ system, the $CaCO_3$ ($a \approx 4.99$, $c \approx 17.08$ Å; space group $R\bar{3}c$, Z = 6) and $CaMg(CO_3)_2$ ($a \approx 4.80$, $c \approx 16.00$ Å; space group $R\bar{3}$, Z = 3) compounds with calcite and dolomite structures, respectively, are known. The $Ca^{2+}$ and $Mg^{2+}$ ions occupy regular and distorted octahedral sites in the calcite and dolomite structures, respectively, with an ordered arrangement of the $Ca^{2+}$ and $Mg^{2+}$ ions along the 3-fold axis, which leads to a decrease in the symmetry of $CaMg(CO_3)_2$ compared to the $CaCO_3$. Based on the transformed compositions ($CaCO_3 \equiv Ca_4(CO_3)_4$, $CaMg(CO_3)_2 \equiv Ca_2Mg_2(CO_3)_4$) and addiction of $Ln^{3+}$ ions to predominantly trigonal-prismatic ($Ln$ = La–Gd) or octahedral ($Ln$ = Tb–Lu; as well as $M^{3+}$ ions) coordination [35], it is possible that compounds with the initial composition $Ln^{3+}M^{3+}_3(BO_3)^{3-}_4$ can have dolomite-like and calcite-like structures with an ordered arrangement of $Ln^{3+}$ and $M^{3+}$ ions over the octahedral sites both separately (full positional ordering) and jointly (partial positional ordering). This can be expected, for example, for the $Ln^{3+}M^{3+}_3(BO_3)^{3-}_4$ with the $Ln$ = Tm ($r_{Tm}^{VI}$ = 0.88 Å) or Yb ($r_{Yb}^{VI}$ = 0.87 Å) in combination with the $M^{3+}$ = Cr ($r_{Cr}^{VI}$ = 0.615 Å) or Ga ($r_{Ga}^{VI}$ = 0.620 Å), for which $\Delta r_{Ln-M}$ = ~0.25 Å, forming the dolomite-type structure ($\Delta r_{Ca-Mg}$ = 0.28 Å), and for the $Ln^{3+}M^{3+}_3(BO_3)^{3-}_4$ with the $Ln$ = Tb ($r_{Tb}^{VI}$ = 0.92 Å) in combination with the $M^{3+}$ = Sc ($r_{Sc}^{VI}$ = 0.745 Å) ($\Delta r_{Tb-Sc}$ = ~0.175 Å), forming the calcite-type structure. Indeed, a polycrystalline sample with the initial composition $Yb^{3+}Ga^{3+}_3(BO_3)^{3-}_4$, which was sintered between 575 and 1050 °C [34], and a single crystal with the initial composition $Tb^{3+}Sc^{3+}_3(BO_3)^{3-}_4$, obtained by the Czochralski method [35,83], crystallize with a decrease in symmetry, but with the same unit cell parameters, forming superstructures to dolomite with the space group $R3$ ($a = 4.726(3)$, $c = 15.43(2)$ Å) and calcite with the space group $R\bar{3}$ ($a = 4.773(5)$, $c = 15.48(1)$ Å), respectively. The XRD analysis of $Tb^{3+}Sc^{3+}_3(BO_3)^{3-}_4$ single crystal allowed to reveal additional diffraction reflections $h\bar{h}0l$ with the $h + l = 2n$, which are absent for the space group $R\bar{3}c$ and possible for the space groups $R\bar{3}$ and $R3$. For the crystals obtained, a non-synchronous second harmonic generation was not observed, which indicates the space groups $R\bar{3}$.

Single crystals with the initial composition $LnSc_3(BO_3)_4$ with the $Ln$ = La [74,75], Ce [81], Pr [80,88], Nd [77,80,88], Sm [80,88], Eu [88], obtained by the flux method (the $NdSc_3(BO_3)_4$ were grown by the Czochralski method [77]), and also with the $Ln$ = Gd [36,37] and Y [36,37], grown by the TSSG method, have a modification with the huntite structure (space group $R32$) (Figures 1 and 2). Fedorova et al. [78] as well as Li et al. [25] and Ye et al. [31] could not obtain a stable modification of $LaSc_3(BO_3)_4$ with the space group $R32$ by solid state reaction ($LaSc_3(BO_3)_4$ samples with the space group $C2/c$ have been

obtained) and by a high-temperature solution method, respectively, suggesting that this phase seems to be metastable or stable in a narrow temperature range. It should be noted that the space group *R*32 for the *Ln*Sc$_3$(BO$_3$)$_4$ with the *Ln* = La [74,75], Ce [81], Pr [88], Nd [77,88], Sm [88], Eu [88] is determined by the single crystal XRD study, whereas for the *Ln* = Gd, (Y), the space group *R*32 is stated in [36,37] without any experimental confirmation, noting only that single crystals grown by the TSSG method exhibited well developed facets having the form of rhombohedral prisms characteristic of the space group *R*32.

For the crystals grown by the Czochralski method from the initial charges with the compositions Pr$_{1.1}$Sc$_{2.9}$(BO$_3$)$_4$ (PSB–1.1) and Pr$_{1.25}$Sc$_{2.75}$(BO$_3$)$_4$ (PSB–1.25) ($\Delta r_{Pr-Sc}$ = 0.245 Å), NdSc$_3$(BO$_3$)$_4$ (NSB–1.0) and Nd$_{1.25}$Sc$_{2.75}$(BO$_3$)$_4$ (NSB–1.25) ($\Delta r_{Nd-Sc}$ = 0.235 Å), a symmetry decrease from the space group *R*32 to *P*321 or *P*3 (Figures 1 and 2) was found by the XRD analysis (it should be noted that the value $\Delta r_{Nd-Sc}$ (Å) is less than the critical value $\Delta r_{Ln-M}$ = ~0.25 Å, at which the derivatives of the huntite structure are formed; $\Delta r_{Pr-Sc}$ (Å) is actually at the stability limit). The extinction laws for an overwhelming number of diffraction reflections witness a crystallization of these compounds in the space group *R*32. However, 60% of the additional reflections with the $I \geq 3\sigma(I)$ are described within the framework of the superstructure having the huntite unit cell parameters, but with the space group *P*321 or *P*3 (the structures were solved in the space group *P*321) with the $h + k = 3n$, $l = 2n + 1$ for *hkl* [82]. In crystal structures with the space group *P*321 (Figure 4) compared with those with the space group *R*32 (Figure 3), the *Ln* and Sc crystallographic sites (Figure 3a–f), are split into two (Figure 4a–f), *Ln*1 and *Ln*2 (Figure 4g), sites with a distorted trigonal-prismatic oxygen environment and two, Sc1 and Sc2 (Figure 4h), sites with a distorted octahedral oxygen environment, respectively.

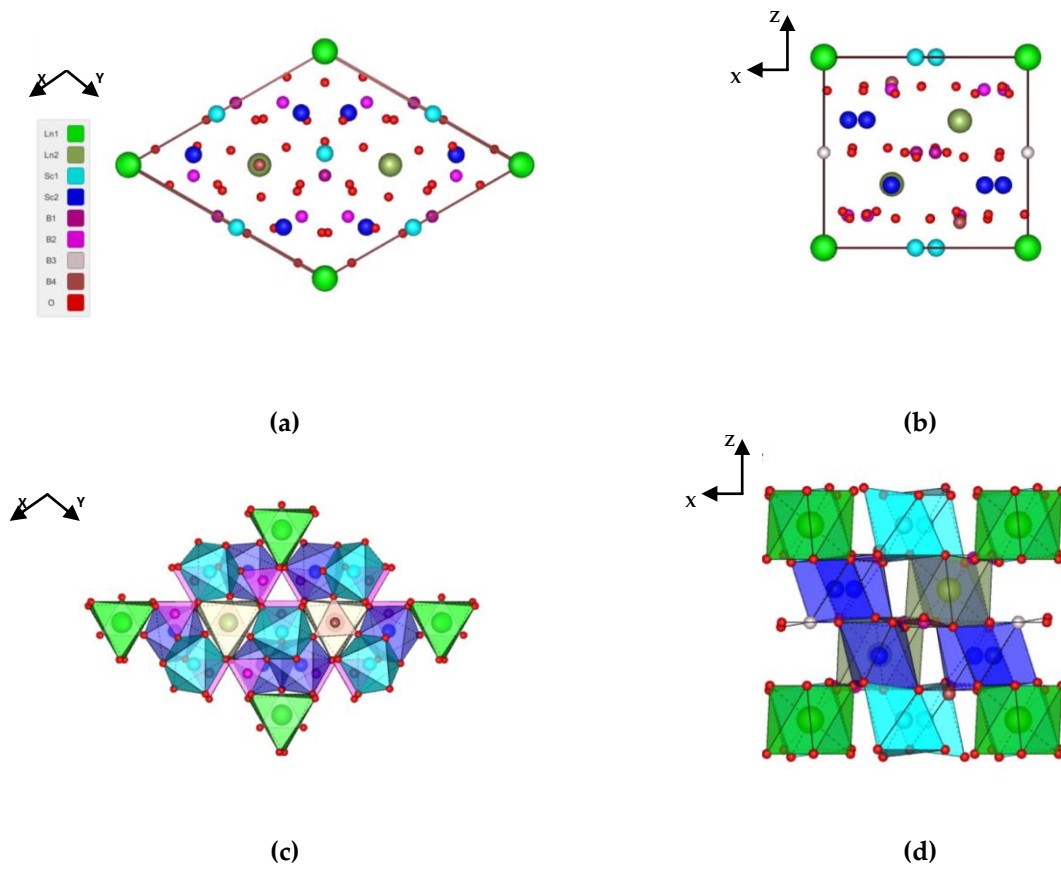

(a)

(b)

(c)

(d)

**Figure 4.** *Cont.*

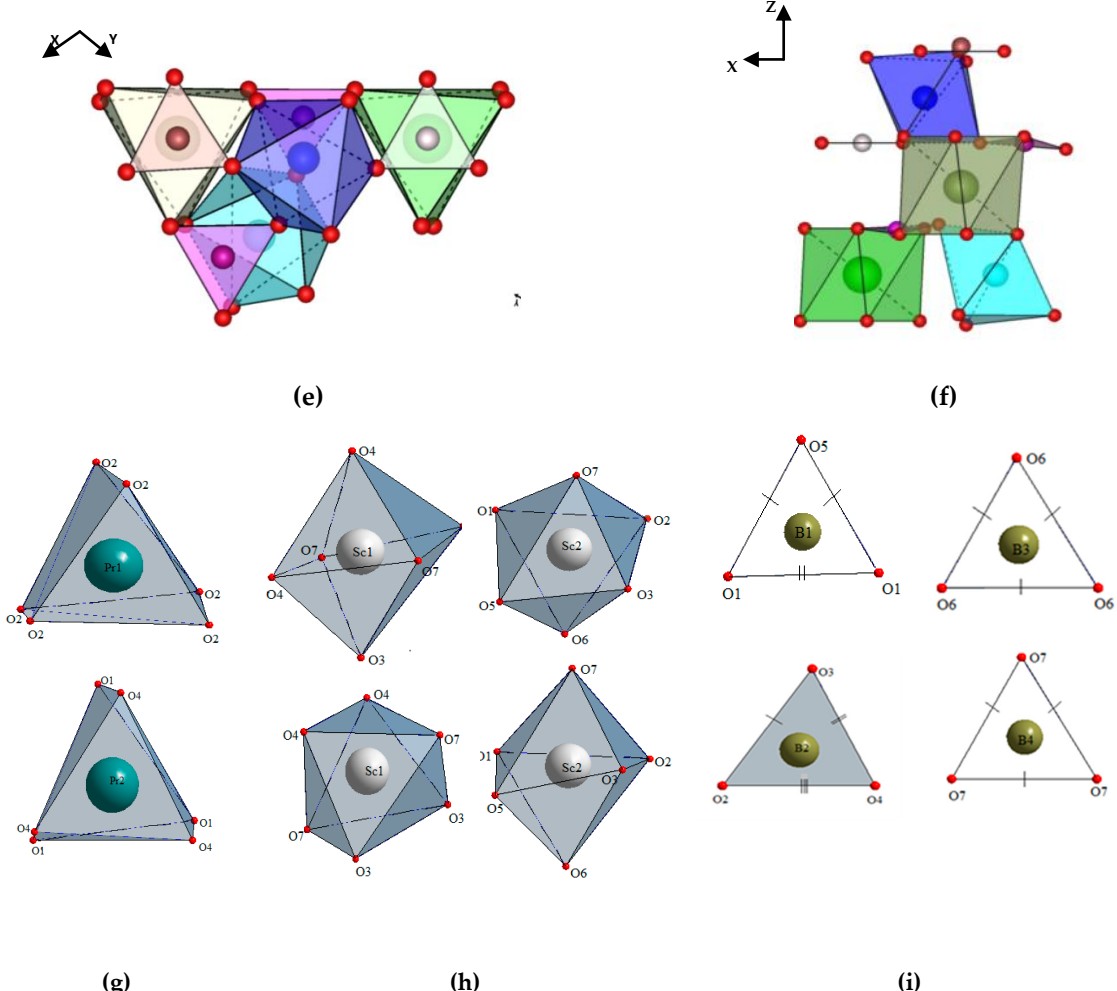

**Figure 4.** The unit cell of the $PrSc_3(BO_3)_4$ (PSB–1.1 and PSB–1.25) structure (space group *P*321) projected onto the (**a**) XY and (**b**) XZ planes; Combination of the coordination polyhedra projected onto the (**c**) XY and (**d**) XZ planes; Combination of selected coordination polyhedra projected onto the (**e**) XY and (**f**) XZ planes; Coordination polyhedra for the (**g**) Pr1 and Pr2, (**h**) Sc1 and Sc2, (**i**) B1–B4.

In addition, the number of B crystallographic sites is also increased in the structure with the space group *P*321 (Figure 4i) compared with the *R*32 one (Figure 3i). A comparison of the XZ projections of the structures with the space groups *R*32 (Figure 3b) and *P*321 (Figure 4b) indicates an alternation of layers of atoms La, Sc–B, O–La, Sc (Figure 3b) and La1, Sc1–B, O–La2, Sc2–B, O (Figure 4b) and corresponding polyhedra (Figure 3d,f and Figure 4d,f) along the Z axis.

The refined crystal compositions can be written as $[(Pr_{0.419}Sc_{0.081(4)})(1)]Pr_{0.5}(2)Sc_3(BO_3)_4$ $((Pr_{0.919}Sc_{0.081(4)})Sc_3(BO_3)_4)$ (PSB–1.1) and $[(Pr_{0.424}Sc_{0.076(4)})(1)]Pr_{0.5}(2)Sc_3(BO_3)_4$ $((Pr_{0.924}Sc_{0.076(4)})Sc_3(BO_3)_4)$ (PSB–1.25) [82], from which it follows that a symmetry decrease is caused by the distribution of (Pr, Sc) "atoms" and Pr atoms over two trigonal-prismatic sites (partial positional ordering) (Figure 4). Results of the XRD analysis with a refinement of positional and atom displacement parameters of single crystals with the initial composition $PrSc_3(BO_3)_4$, obtained by the flux method, are given in [88]. On the basis of systematic absences *hkil*: $-h + k + l \neq 3n$ and a successful refinement of the data for crystal, the space group was determined to be *R*32; the real composition of the crystal (i.e., the refinement of the site occupancies) was not determined [88].

Crystals with the initial compositions $NdSc_3(BO_3)_4$ (NSB–1.0) and $Nd_{1.25}Sc_{2.75}(BO_3)_4$ (NSB–1.25) with the space group *P*321 (in Figures 1 and 2, it is given as *P*321*) have the refined compositions $NdSc_3(BO_3)_4$ and $(Nd_{0.500}(1)[Nd_{0.410}Sc_{0.090(20)}(2)]Sc_3(BO_3)_4$ $((Nd_{0.910}Sc_{0.090(20)})Sc_3(BO_3)_4)$, respectively.

These structures differ from each other (Figures 5 and 6) by an additional presence of the Sc ions in the trigonal-pyramidal Nd2 sites in the NSB–1.25 structure. It should be noted that the NSB structure is represented by right (NSB–1.0) (Figure 5a–f) and left (NSB–1.25) (Figure 6a–f) forms.

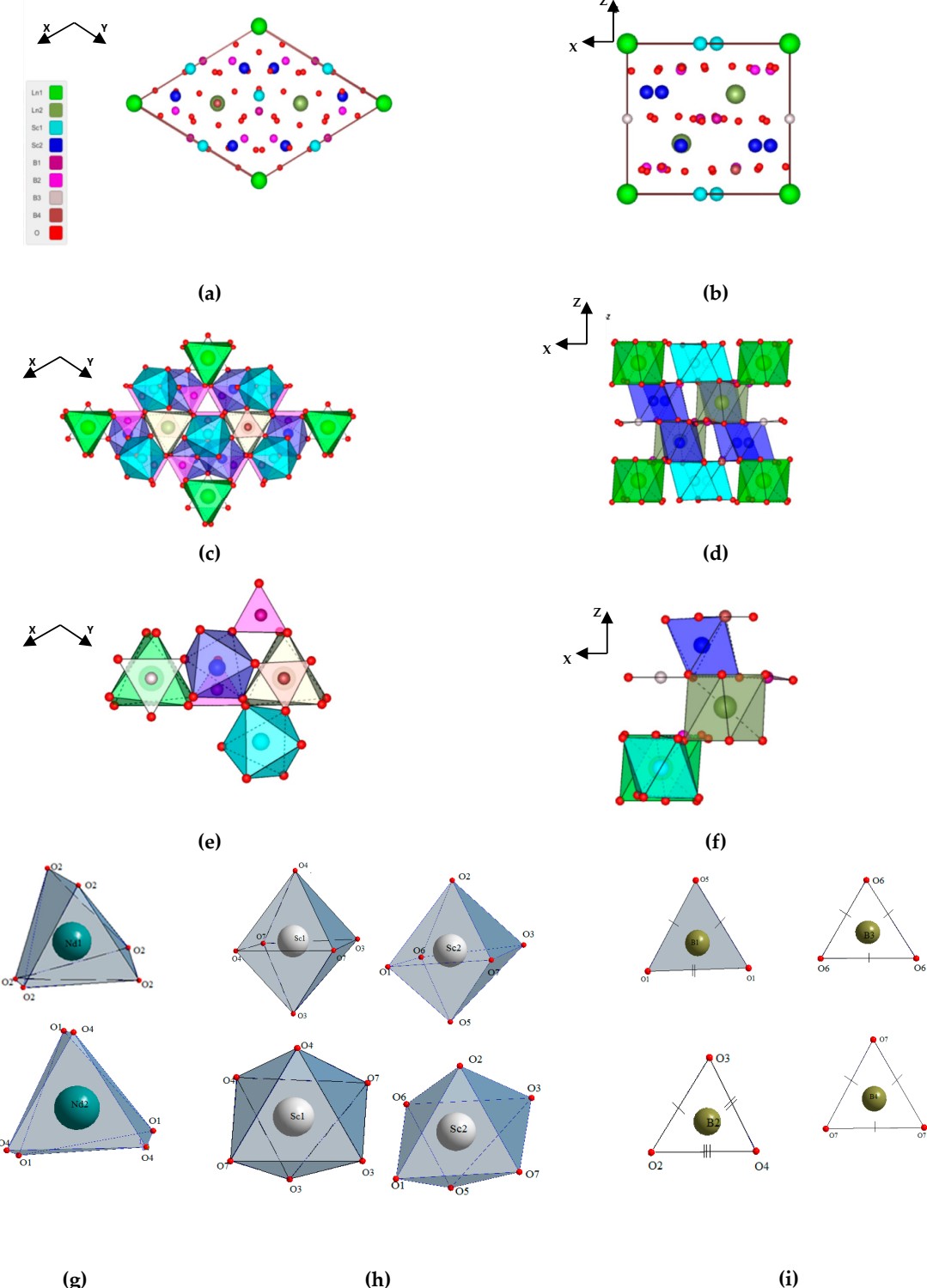

**Figure 5.** The unit cell of the $NdSc_3(BO_3)_4$ (NSB–1.0) structure (space group *P*321) projected onto the (**a**) XY and (**b**) XZ planes; Combination of the coordination polyhedra projected onto the (**c**) XY and (**d**) XZ planes; Combination of selected coordination polyhedra projected onto the (**e**) XY and (**f**) XZ planes; Coordination polyhedra for the (**g**) Nd1 and Nd2, (**h**) Sc1 and Sc2, (**i**) B1–B4.

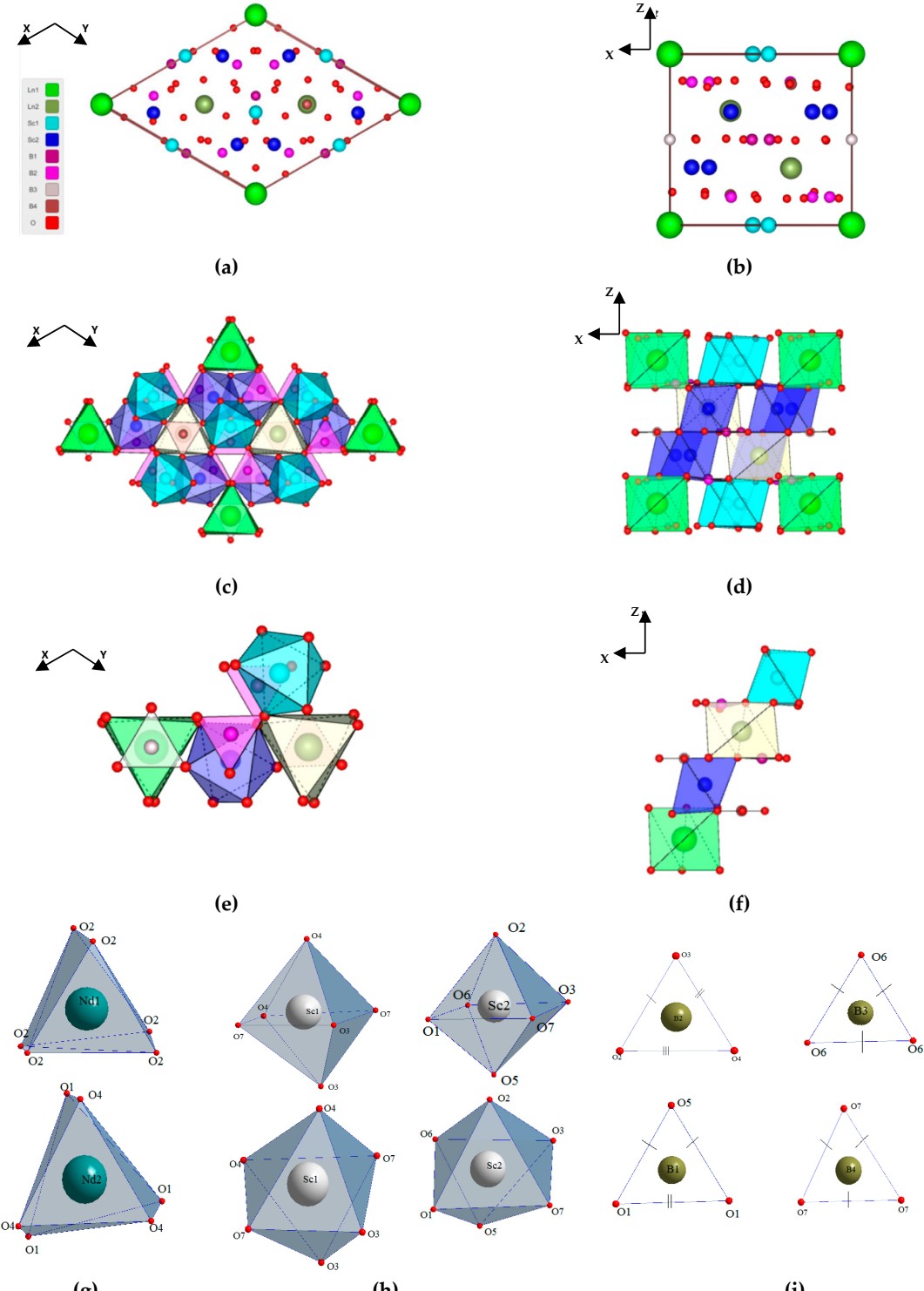

**Figure 6.** The unit cell of the NdSc$_3$(BO$_3$)$_4$ (NSB–1.25) structure (space group *P*321) projected onto the (**a**) XY and (**b**) XZ planes; Combination of the coordination polyhedra projected onto the (**c**) XY and (**d**) XZ planes; Combination of selected coordination polyhedra projected onto the (**e**) XY and (**f**) XZ planes; Coordination polyhedra for the (**g**) Nd1 and Nd2, (**h**) Sc1 and Sc2, (**i**) B1–B4.

The NdO$_6$ coordination polyhedra in the NSB–1.0 structure (Figure 5g) are more distorted than those in the NSB–1.25 one (Figure 6g), and the ScO$_6$ polyhedra in the NSB–1.0 structure (Figure 5h) are more regular (especially the Sc1O$_6$ polyhedron) than those in the NSB–1.25 one (Figure 6h). It confirms,

firstly, a difference in their real compositions and, secondly, a 'stress removal' in the NSB–1.25 structure due to the presence of Sc atoms in trigonal-prismatic polyhedra.

Compared to the PSB structures (Figure 4), in the NSB–1.0 structure (Figure 5), the $ScO_6$ coordination polyhedron (Figures 4h and 5h) and one B polyhedron (Figures 4i and 5i) are less distorted, as well as the $\varphi$ rotation angle in the $Ln2O_6$ coordination polyhedron is smaller (Figures 4g and 5g), which is caused by the presence of Sc atoms in the Pr site in the PSB structures.

As a result of the analysis of Figures 5 and 6, it can be concluded that the crystals grown by the Czochralski method from the charges with compositions $NdSc_3(BO_3)_4$ (NSB–1.0) and $Nd_{1.25}Sc_{2.75}(BO_3)_4$ (NSB–1.25) are characterized by the different structure disordering compared to the $Pr_{1.1}Sc_{2.9}(BO_3)_4$ (PSB–1.1) and $Pr_{1.25}Sc_{2.75}(BO_3)_4$ (PSB–1.25); NSB and PSB are isotypic structures (space group $P321$) (Figures 4–6). The differences are related to a distribution of the *Ln* and Sc ions over the trigonal-prismatic sites, resulting in another type of structure, namely, unequal $Nd1O_6$ and $Nd2O_6$ trigonal prisms in the NSB compared to the PSB structures and a different character of the changes in the interatomic distances in these polyhedra, primarily, the Nd(Pr)2–O4 и Nd(Pr)2–O1 ones [82]. The differences in the PSB–1.1, PSB–1.25, NSB–1.0, NSB–1.25 structures can be traced, for example, on their XZ projections (Figures 4b, 5b and 6b), paying attention to the B and O atoms.

Depending on the composition and growth conditions, the huntite-family compounds have monoclinic modifications (Figures 1 and 2) with the centrosymmetric space group $C2/c$ (Figure 7) and non-centrosymmetric space groups $Cc$ (the extinction laws are the same as those for the space group $C2/c$) (Figure 8) and $C2$ (Figure 9) (the extinction laws are the same as those for the space groups $C2/m$, $Cm$).

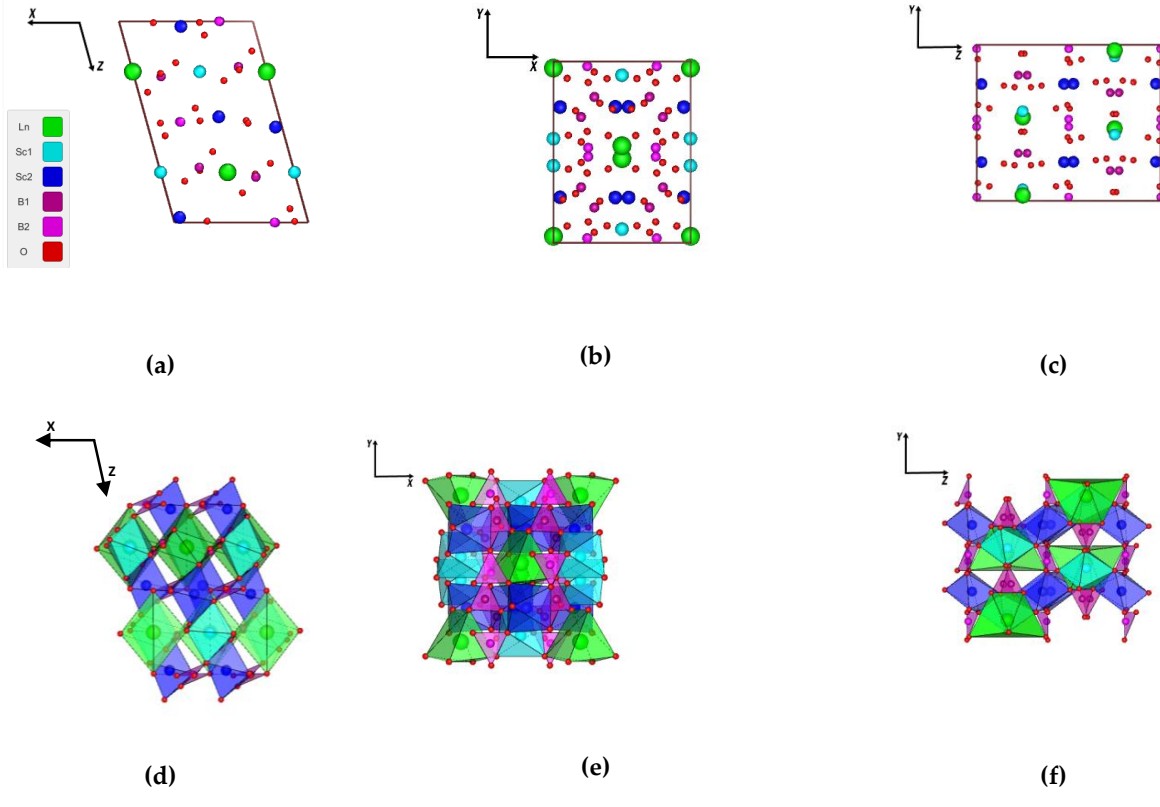

**Figure 7.** *Cont.*

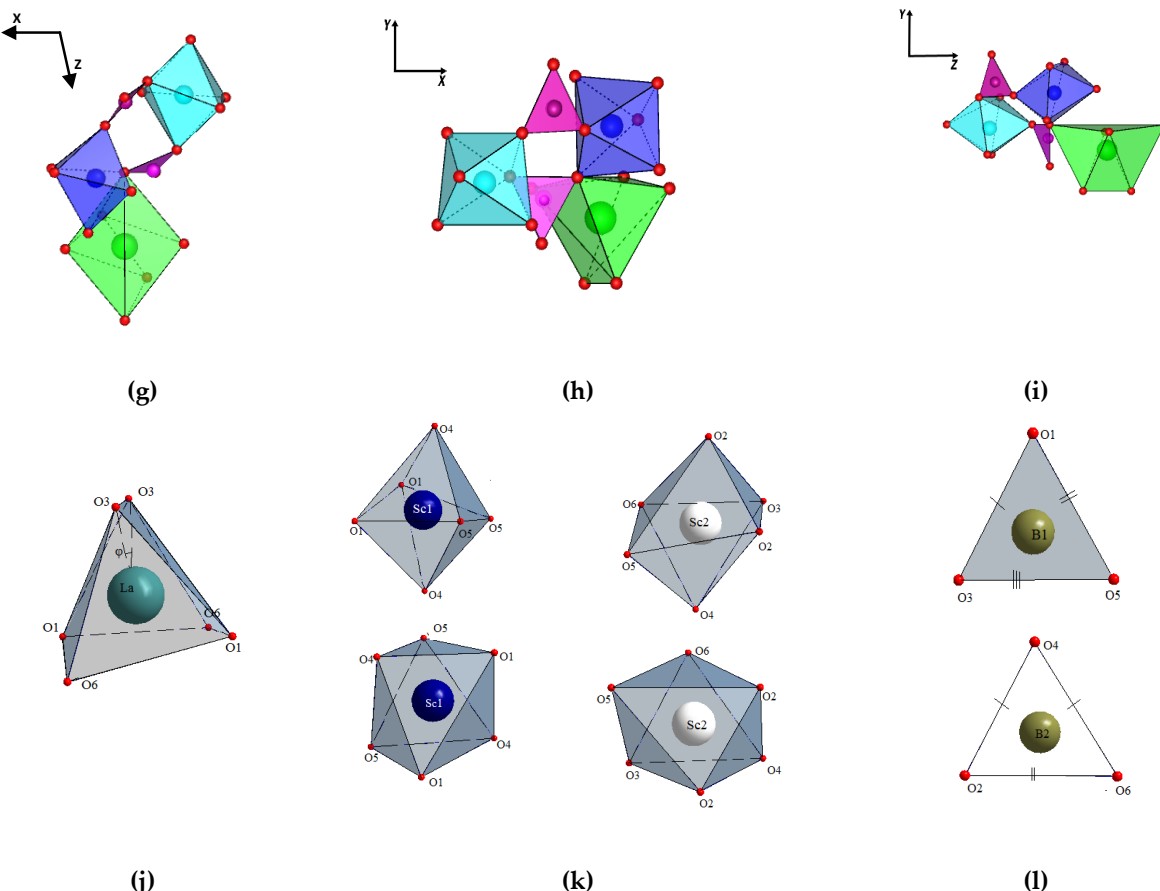

**Figure 7.** The unit cell of the *LnM*$_3$(BO$_3$)$_4$ structure (space group *C2/c*) projected onto the (**a**) XZ, (**b**) XY, (**c**) YZ planes; Combination of the coordination polyhedra projected onto the (**d**) XZ, (**e**) XY, (**f**) YZ planes; Combination of selected coordination polyhedra projected onto the (**g**) XZ, (**h**) XY, (**i**) YZ planes; Coordination polyhedra for the (**j**) *Ln*, (**k**) *M1* and *M2*, (**l**) B1 and B2.

The space group *C2/c* is known for the *Ln*Al$_3$(BO$_3$)$_4$ with the *Ln* = Pr, Nd, Sm, Eu, Tb, Ho (the structure is refined, for example, in [97]); *Ln*Fe$_3$(BO$_3$)$_4$ with the *Ln* = Nd; *Ln*Cr$_3$(BO$_3$)$_4$ with the *Ln* = La, Pr, Nd, Sm, Tb, Dy, Ho; *Ln*Sc$_3$(BO$_3$)$_4$ with the *Ln* = La, Ce, Pr (Table 1). The centrosymmetry was determined by spectroscopic studies using a factor-group analysis of vibrations (for the *Ln*Al$_3$(BO$_3$)$_4$ with the *Ln* = Pr [42], Nd [42,45]; *Ln*Fe$_3$(BO$_3$)$_4$ with the *Ln* = Nd [45]; *Ln*Cr$_3$(BO$_3$)$_4$ with the *Ln* = La (Figure 10), Pr, Tb, Dy, Ho [70,71], Nd [26,45,70,71], Sm [70,72]). The *Ln*Sc$_3$(BO$_3$)$_4$ crystals with the *Ln* = La, Ce, obtained by the Czochralski method, crystallize in the centrosymmetric space group *C2/c*), according to a study of their nonlinear optical properties [35]. The XRD investigation of the CeSc$_3$(BO$_3$)$_4$ microcrystal with the space group *C2/c* indicates the similarity of charge and as-grown crystal compositions [82].

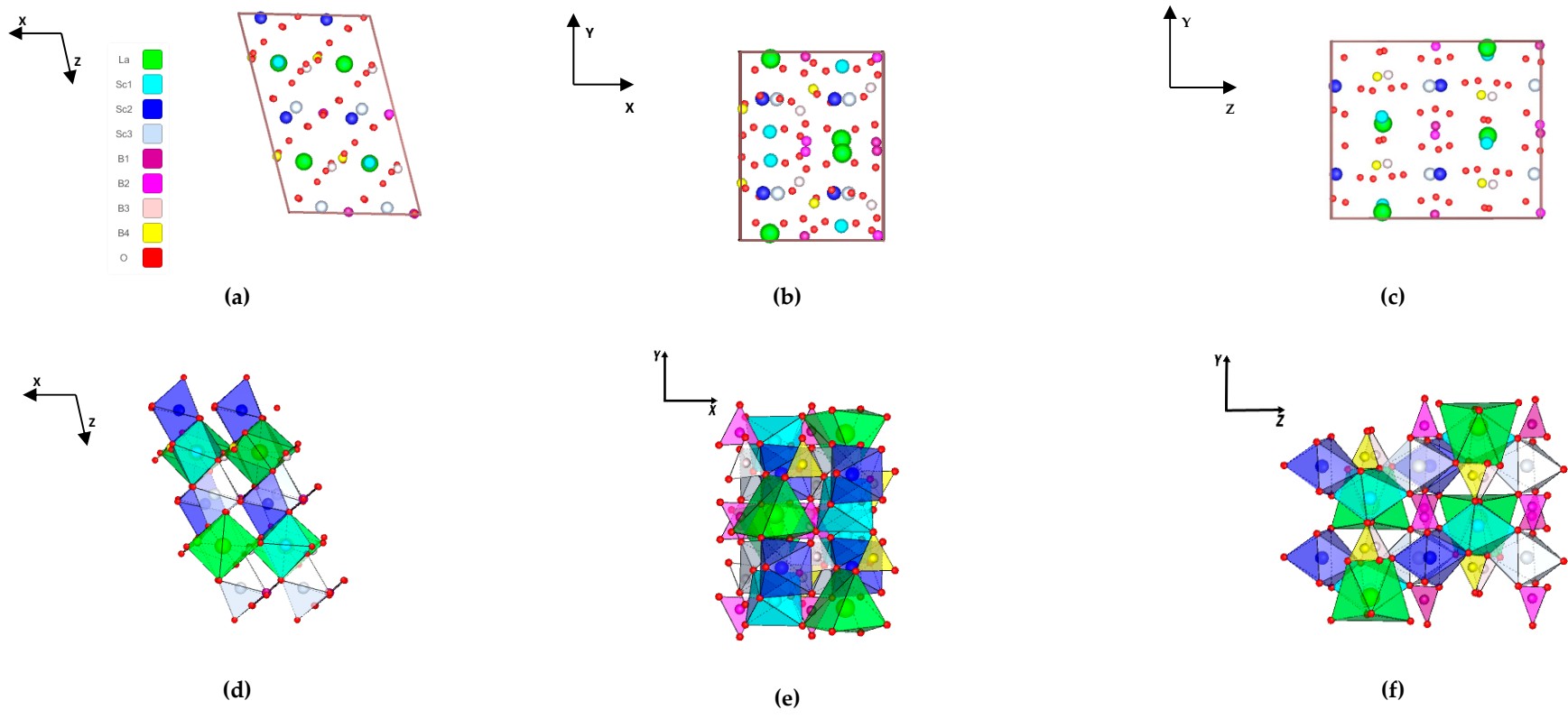

**Figure 8.** *Cont.*

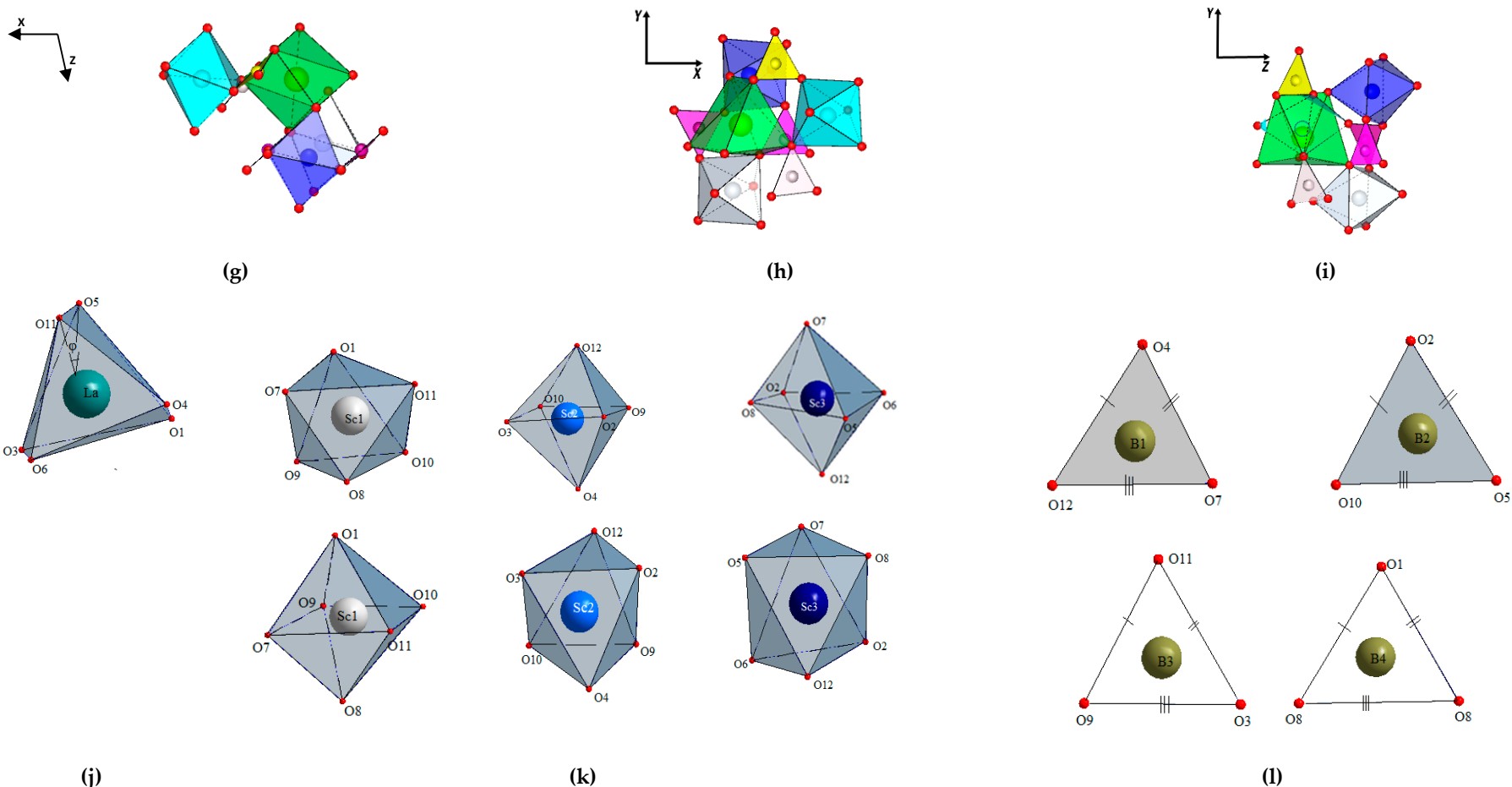

**Figure 8.** The unit cell of the $LnM_3(BO_3)_4$ structure (space group *Cc*) projected onto the (**a**) XZ, (**b**) XY, (**c**) YZ planes; Combination of the coordination polyhedra projected onto the (**d**) XZ, (**e**) XY, (**f**) YZ planes; Combination of selected coordination polyhedra projected onto the (**g**) XZ, (**h**) XY, (**i**) YZ planes; Coordination polyhedra for the (**j**) *Ln*, (**k**) *M1–M3*, (**l**) B1–B4.

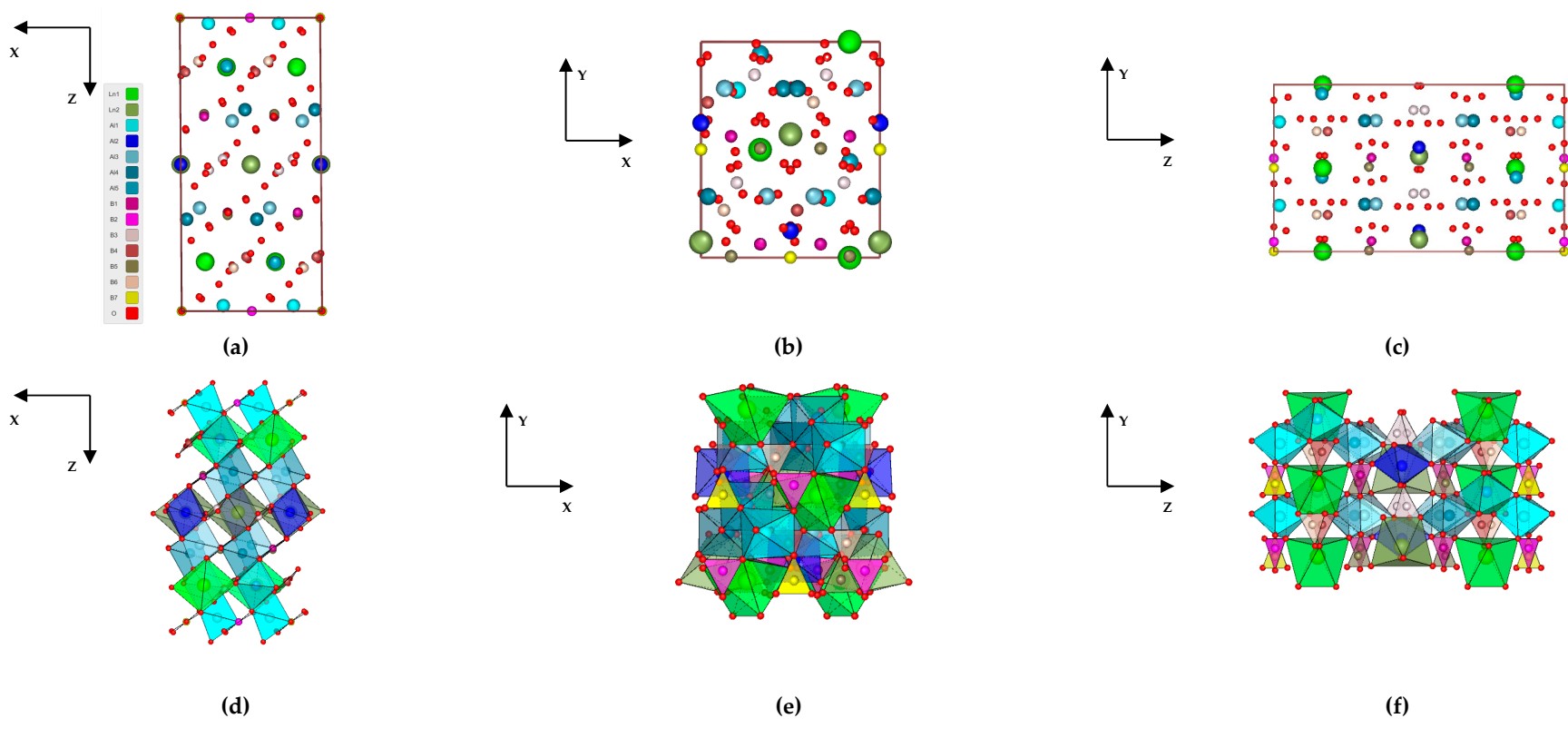

**Figure 9.** *Cont*.

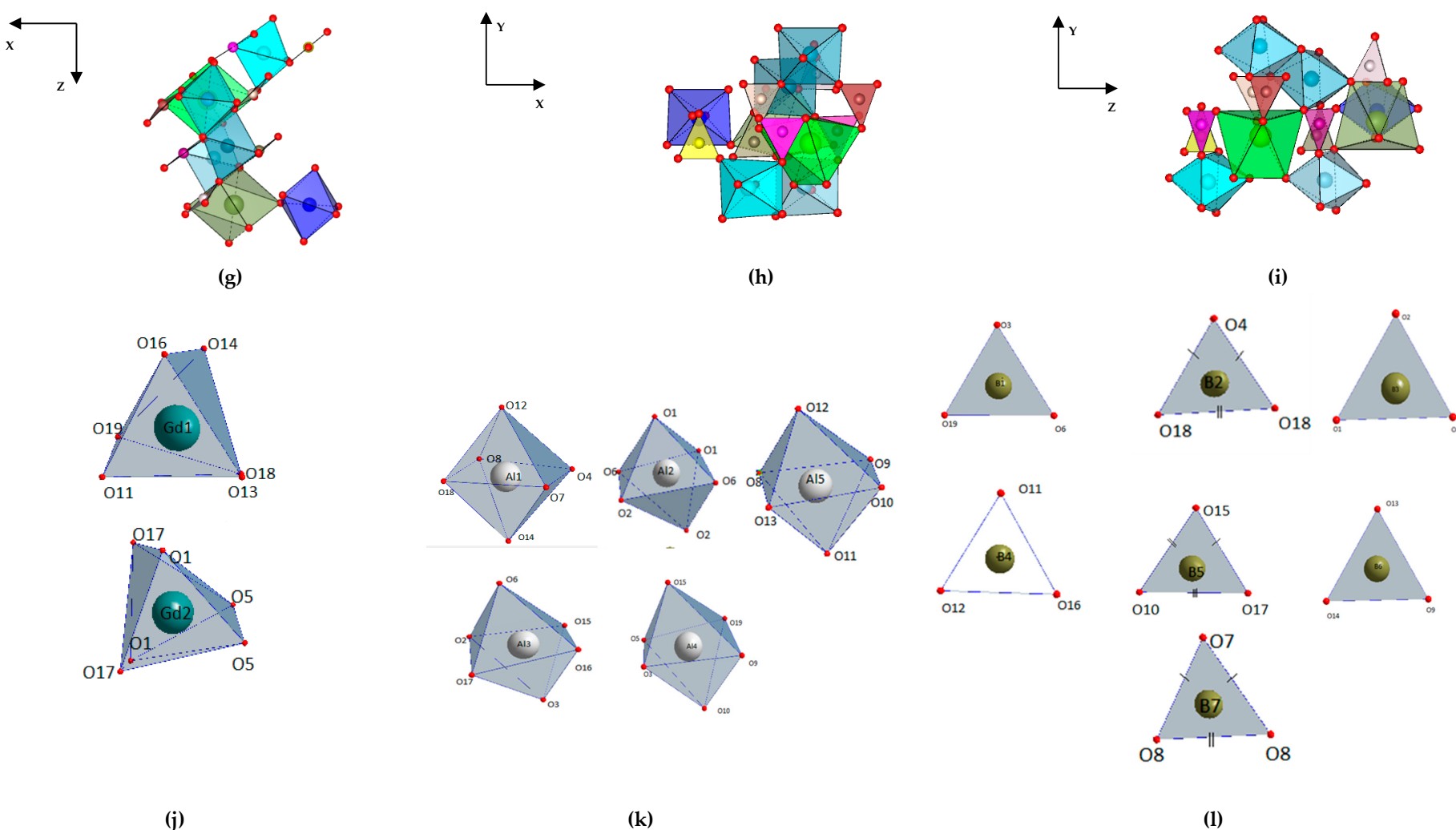

**Figure 9.** The unit cell of the $LnM_3(BO_3)_4$ structure (space group C2) projected onto the (**a**) XZ, (**b**) XY, (**c**) YZ planes; Combination of the coordination polyhedra projected onto the (**d**) XZ, (**e**) XY, (**f**) YZ planes; Combination of selected coordination polyhedra projected onto the (**g**) XZ, (**h**) XY, (**i**) YZ planes; Coordination polyhedra for the (**j**) *Ln1* and *Ln2*, (**k**) *M1–M5*, (**l**) B1–B7.

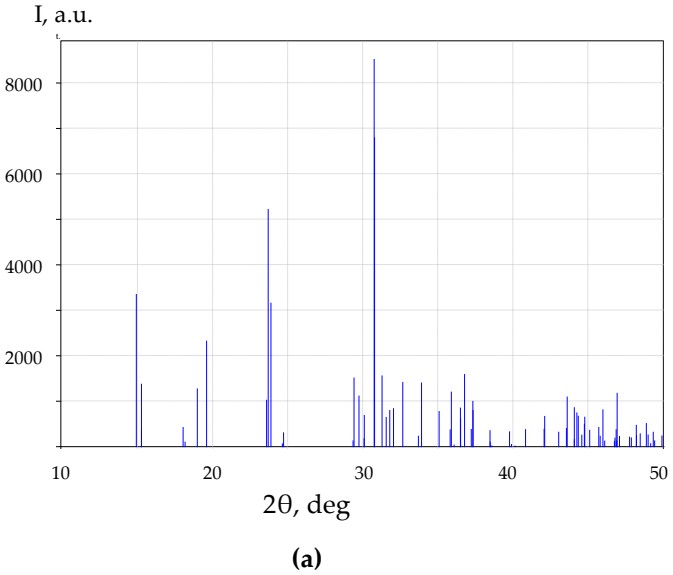

**(a)**

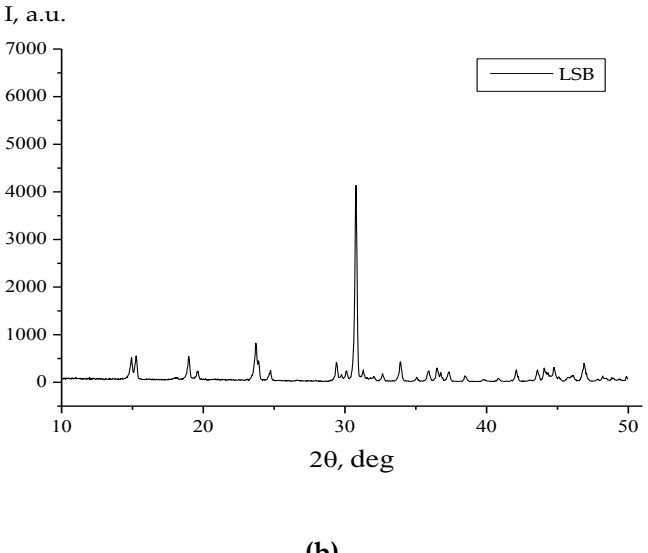

**(b)**

**Figure 10.** (**a**) Theoretical and (**b**) experimental diffraction patterns for the α - LaSc$_3$(BO$_3$)$_4$ structure (space group *C*2/*c*).

In the crystal structure with the space group *C*2/*c* (Figure 7) the *Ln* atom is located at the center of a distorted trigonal prism (CN *Ln* = 2 + 2 + 2) (Figure 7j); the upper triangular face of prism is rotated with respect to the lower one by an angle φ = 8°. The Sc1 and Sc2 atoms are located in distorted octahedra (CN Sc1 = 2 + 2 + 2, CN Sc2 = 1 + 1 + 1 + 1 + 1 + 1) (Figure 7k), and the B1 and B2 atoms are at the centers of scalene triangles (CN B1 = 1 + 1 + 1; CN B2 = 1 + 1 + 1) (Figure 7l). A comparison of crystal structures with the space groups *R*32 (Figure 3) and *C*2/*c* (Figure 7) shows their similarity (the same set of polyhedra; however, in the structure with the space group *C*2/*c*, there are two crystallochemically-different Sc atoms rather that one as in the structure with the space group *R*32) (Figure 3g,h,I and Figure 7j,k,l), on the one hand, and differences (the polyhedra in the structure with the space group *C*2/*c* are more distorted), on the other hand, i.e., the structure with the space group *C*2/*c* is more disordered compared to the *R*32. A similar alternation of layers of atoms ... Ln, Sc–B, O–Ln, Sc and Ln, Sc2–B, O–Sc1–B, O–Ln, Sc1, Sc2–B, O in the structures with the space groups

*R*32 (XZ projection; Figure 3b) and *C*2/*c* (XZ projection; Figure 7a) and the corresponding polyhedra (Figure 3d,f and Figure 7d,g) should be mentioned.

The low temperature phase $\gamma$-LaSc$_3$(BO$_3$)$_4$ crystallizes in the space group *Cc* ($a$ = 7.740(3), $b$ = 9.864(2), $c$ = 12.066(5) Å, $\beta$ = 105.48(5)°) (Figure 8) [79], but there is no information on how its noncentrosymmetry was determined. The $\gamma$-LaSc$_3$(BO$_3$)$_4$ single crystal was grown by the TSSG method from a flux and its structure was solved by direct method and refined by full-matrix least squares without any refinement of real crystal composition [79]. In the structure with the space group *Cc* (Figure 8), which is positionally and structurally disordered compared with the $\alpha$- LaSc$_3$(BO$_3$)$_4$ modification with the space group *C*2/*c* (Figure 7) the Sc1, Sc2, and Sc2 atoms are located in distorted octahedra (CN Sc = 1 + 1 + 1 + 1 + 1 + 1) (Figure 8k), and the B1, B2, B3, and B4 atoms are at the centers of scalene triangles (CN B = 1 + 1 + 1) (Figure 8l). According to Wang et al. [79], in the crystal structure of $\gamma$- LaSc$_3$(BO$_3$)$_4$ (space group *Cc*), the La atoms are at the centers of distorted octahedra with the CN La = 1 + 1 + 1 + 1 + 1 + 1. However, according to the rotation angle $\varphi$ = 14° between the upper and lower faces (Figure 8j), which is larger than that in the structure with the space group *C*2/*c* (Figure 7j), the La polyhedron is a trigonal prism, since it is nominally considered that a polyhedron with the CN = 6 is a distorted trigonal prism or a distorted octahedron in the angle ranges 0° < $\varphi$ <<~ 30° or ~30° < $\varphi$ < 60°, respectively. Actually, based on a structure with the space group *C*2/*c*, the Sc2 crystallographic site (Figure 7a–i) is 'split' into two sites, Sc2 and Sc3 (Figure 8a–i), the B1 site (Figure 7l) is 'split' into the B1 and B2 sites (Figure 8l), and the B2 site (Figure 7l) is 'split' into the B3 and B4 sites (Figure 8l) with the formation of a structure with the space group *Cc*. Hence, a transition from the centrosymmetric structure with the space group *C*2/*c* to non-centrosymmetric structure with the space group *Cc* is obliged to 'split' sites (see, for example, Figures 7f and 8f). In the structures with the space groups *C*2/*c* and *Cc*, a similar alternation of layers of atoms and polyhedra with these atoms is observed on XZ projections: Ln, Sc2–B, O–Sc1–B, O–Ln, Sc1, Sc2–B, O (Figure 7a,d) and Ln, Sc1, Sc2–B, O–Ln, Sc2, Sc3–B, O–Ln, Sc2, Sc3–B, O–Ln, Sc1, Sc3–B, O (Figure 8a,d).

Fedorova et al. [78], based on a complete correspondence (the space groups *C*2/*c* and *Cc* belong to the same diffraction symmetry group with the same extinction laws) of the diffraction patterns of powdered $\alpha$-LaSc$_3$(BO$_3$)$_4$ (space group *C*2/*c*) and $\gamma$- LaSc$_3$(BO$_3$)$_4$ (space group *Cc*), obtained by solid state synthesis and heat-treated at temperatures from 1000 to 1350 °C, concluded that the LaSc$_3$(BO$_3$)$_4$ crystallizes in the space group *C*2/*c*. The experiment described does not allow choosing one of two space groups correctly.

The unit cell parameters of single crystal grown from the charge having composition (Pr$_{1.1}$Sc$_{2.9}$)(BO$_3$)$_4$ ($a$ = 7.7138(6), $b$ = 9.8347(5), $c$ = 12.032(2) Å, $\beta$ = 105.38(7)°) [83] are similar to those for the LaSc$_3$(BO$_3$)$_4$ and CeSc$_3$(BO$_3$)$_4$ ($a$ = 7.7297(3), $b$ = 9.8556(3), $c$ = 12.0532(5)Å, $\beta$ = 105.405(3)°) with the space group *C*2/*c* [82]. The extinction laws for the overwhelming number of diffraction reflections correspond to the space group *C*2/*c* or *Cc* ($h + k = 2n$ for *hkl*, $h = 2n$, $l = 2n$ for *h0l*, $k = 2n$ for *0kl*, $h + k$ for *hk0*, $h = 2n$ for *h00*, $k = 2n$ for *0k0*, $l = 2n$ for *00l*). Non-synchronous second harmonic generation was observed for these crystals, which indicates a non-centrosymmetric space structure *Cc*. A small amount of additional reflections with the $I \geq 3\sigma(I)$, typical for the space groups *C*2/*m*, *C*2 or *Cm* ($h + k = 2n$ for *hkl*, $h = 2n$ for *h0l*, $k = 2n$ for *0kl*, $h + k$ for *hk0*, $h = 2n$ for *h00*, $k = 2n$ for *0k0*), most likely, space group *C*2 as a subgroup of *C*2/*c*, is observed.

The space group *Cc* was not found for the *LnM*$_3$(BO$_3$)$_4$ with the *M* = Al, Fe, Cr, Ga. For the *Ln*Al$_3$(BO$_3$)$_4$ with the *Ln* = Pr, Nd, Eu, Gd, a modification with the space group *C*2 ($a$ = 7.262(3), $b$ = 9.315(3), $c$ = 16.184(8)Å, $\beta$ = 90.37° for the GdAl$_3$(BO$_3$)$_4$) [47] was revealed. In the crystal structure of *Ln*Al$_3$(BO$_3$)$_4$ with the space group *C*2 (Figure 9), five crystallochemically-different Al atoms are located at distorted octahedra (CN Al1, Al3, Al4, Al5 = 1 + 1 + 1 + 1 + 1 + 1, CN Al2 = 2 + 2 + 2) (Figure 9k) and seven crystallochemically-different B atoms have a trigonal environment, two of which are isosceles triangles and the rest ones are scalene (Figure 9l) [41]. The *Ln* (Gd) atoms occupy distorted trigonal prisms with the CN Ln1 = 1 + 1 + 1 + 1 + 1 + 1 and CN Ln2 = 2 + 2 + 2, and the rotation angle is $\varphi$ ~ 20° (Figure 9j), i.e., significantly higher than that for other modifications of huntite-like structures.

This feature of the $LnAl_3(BO_3)_4$ crystal structure with the space group *C*2, which is the most disordered among all the huntite-like structures described (Figure 9a–i), does not exclude a tendency to reorganize a trigonal prism into an octahedron, accompanied by a decrease in the polyhedron volume and degree of its filling [98,99]. It may be the reason or one of the reasons for an absence of this structure for the $LnSc_3(BO_3)_4$ compounds. Nevertheless, a comparison of monoclinic compounds with the space groups *C*2/*c* (Figure 7), *Cc* (Figure 8), and *C*2 (Figure 9) indicates their identical structural features: 'layers' of atoms and polyhedra (Figure 7a,d, Figure 8a,d, Figure 9a,d), a connection of the $LnO_6$ trigonal prisms and ScO6 octahedra through the $BO_3$ triangles (Figure 7g–i, Figure 8g–i, Figure 9g–i). The same structural features can also be observed for the compounds with the space groups *R*32 (Figure 3) and *P*321 (Figures 4–6).

Based on the experimental data available to date, it is possible to limit the region of stability of the phases with the general composition $LnM_3(BO_3)_4$, having the huntite-like structures, as a correlation between the type of *Ln* ion and the difference between the ionic radii of *Ln* and *M* ions (Figure 11).

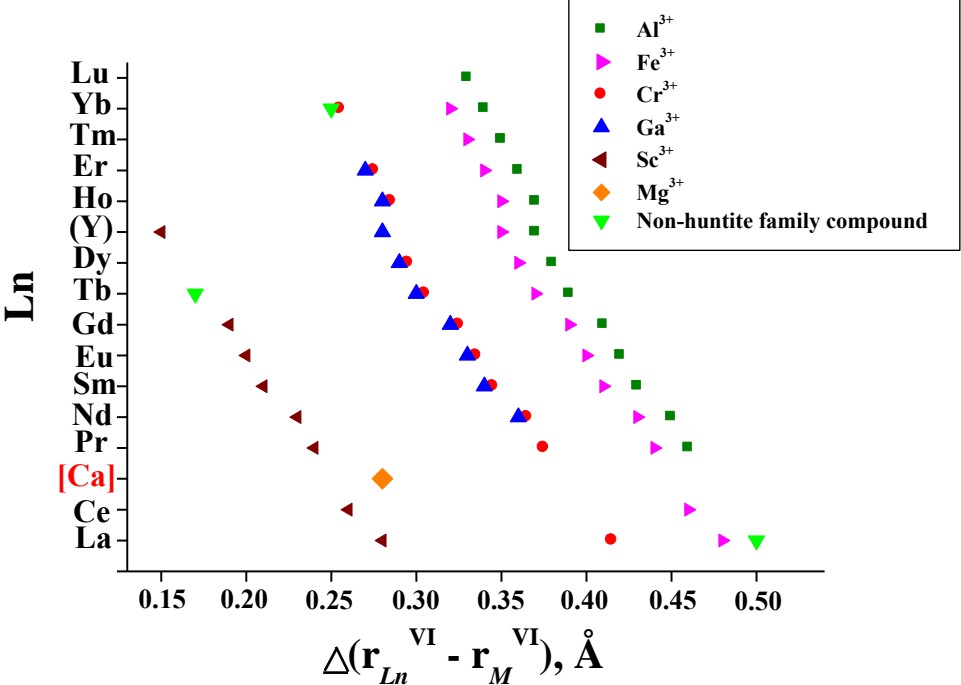

**Figure 11.** The stability region for the phases with the general composition $LnM_3(BO_3)_4$ with the $M^{3+}$ = Al, Fe, Cr, Ga, Sc given as correlation between the type of *Ln* ion and the difference between the ionic radii of *Ln* and *M* ions.

In the coordinates given in Figure 11, the huntite $CaMg_3(CO_3)_4$ (space group *R*32) is located closer to the $LnSc_3(BO_3)_4$, which may explain a formation of superstructures with full or partial ordering of atoms over the crystallographic sites (positional ordering) accompanied by a symmetry decrease in both $CaCO_3$-$MgCO_3$ and $Ln_2O_3$-$Sc_2O_3$-$B_2O_3$ system. In addition, there are other structural features of rare-earth scandium orthoborates.

For polycrystalline and single-crystal samples with the general composition $LnSc_3(BO_3)_4$, for which the $r_{Ln}^{VI}$-$r_{Sc}^{VI}$ (Å) value is in the range of 0.285 (*Ln* = La) − 0.155 (*Ln* = Y) (i.e., with the minimum $\Delta r_{Ln-M}$ (Å) values among all the huntite-like structures) (Figure 11), a probability of obtaining compounds with the stoichiometric composition $LnSc_3(BO_3)_4$ decreases, starting with the *Ln* = Nd ($\Delta r_{Nd-Sc}$ = 0.235 Å). A distinctive feature of samples with the *Ln* = Sm ($\Delta r_{Sm-Sc}$ = 0.215 Å), Eu ($\Delta r_{Eu-Sc}$ = 0.215 Å), and Gd ($\Delta r_{Gd-Sc}$ = 0.195 Å) should be a formation of internal solid solutions in the form $(Ln,Sc)Sc_3(BO_3)_4$ (matrix ions are redistributed over different crystallographic sites resulting in a formation of antisite defects). In samples with the *Ln* from Tb ($\Delta r_{Tb-Sc}$ = 0.175 Å)

to Lu ($\Delta r_{\text{Lu-Sc}}$ = 0.110 Å), internal solid solutions in the form ($Ln$,Sc)(Sc,$Ln$)$_3$(BO$_3$)$_4$ will probably be formed. A formation of internal solid solutions can lead to a decrease in symmetry due to a positional disordering of the $Ln$ and Sc cations, which may ultimately results in the compounds going beyond the limits of the stability region of the huntite-family phases (Figure 11).

The data on the symmetry of compounds with the general composition $Ln$Sc$_3$(BO$_3$)$_4$ are contradictory, in particular, for the modifications with the space group $R$32. A difficulty of revealing the space group $P$321, which was found for the PSB and NSB as a result of a comparison the experimental diffraction patterns with those given in databases and by the refinement of crystal structures of polycrystalline samples or powdered single crystals by the full-profile method (it is usually used in practice), is associated with the almost complete similarity of theoretical diffraction patterns for the space groups $R$32 and $R$321 (Figure 12a,b). The differences relate only to the intensities of individual reflections, shown in Figure 12a,b in a circle.

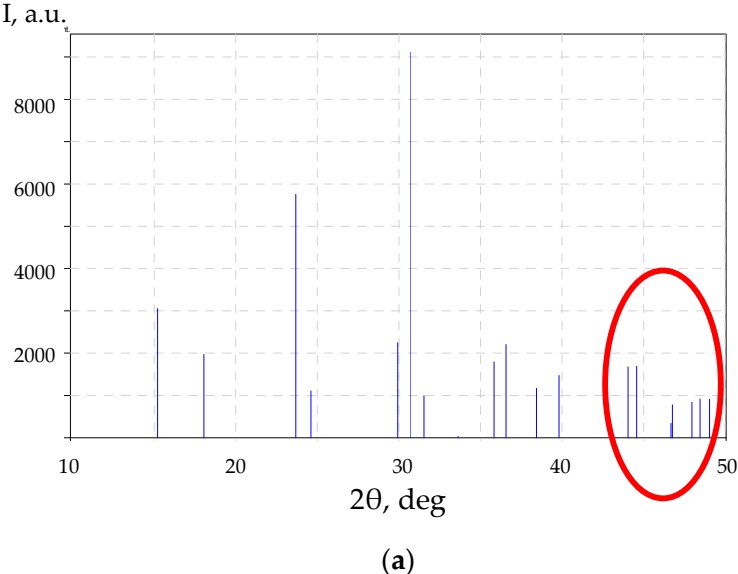

(**a**)

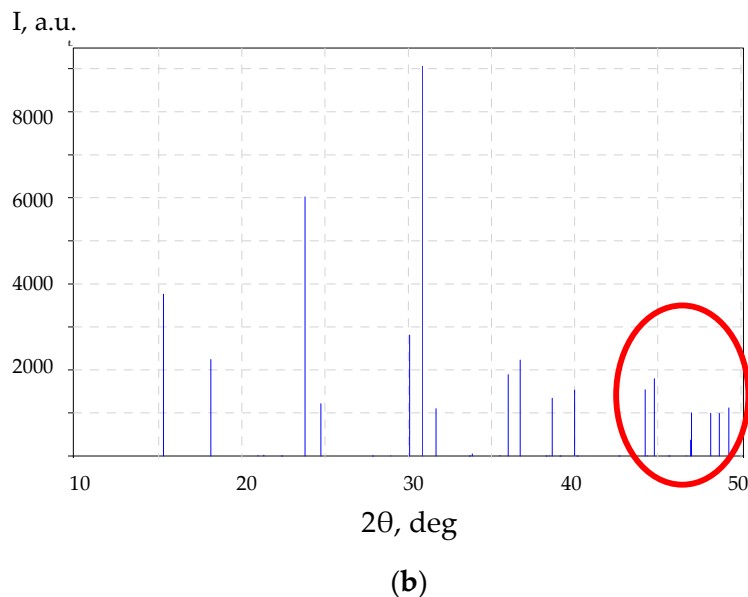

(**b**)

**Figure 12.** *Cont.*

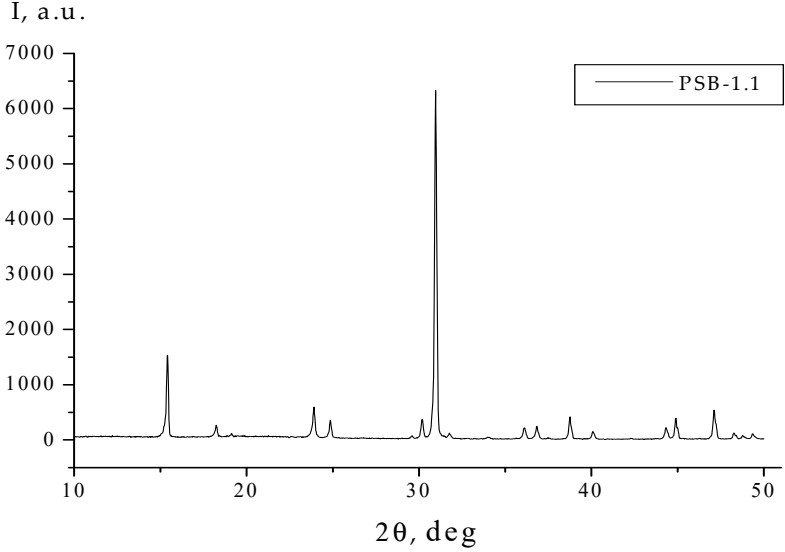

**(c)**

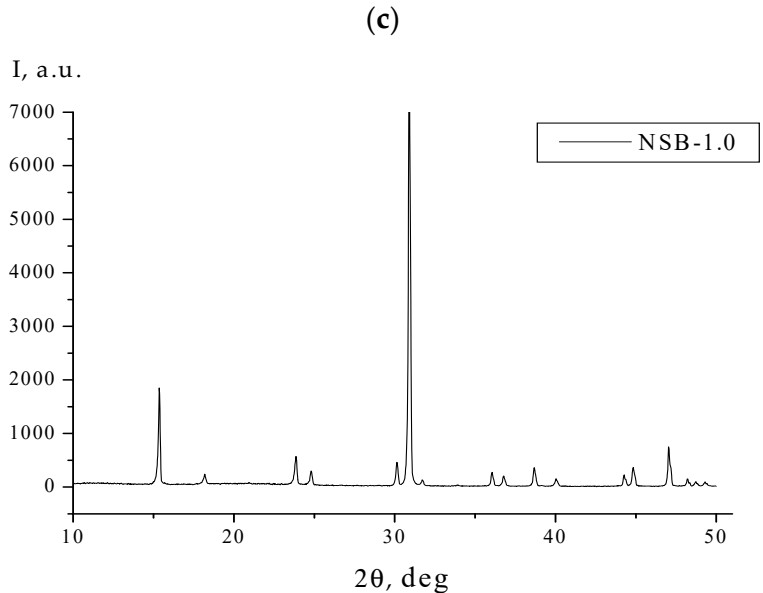

**(d)**

**Figure 12.** *Cont.*

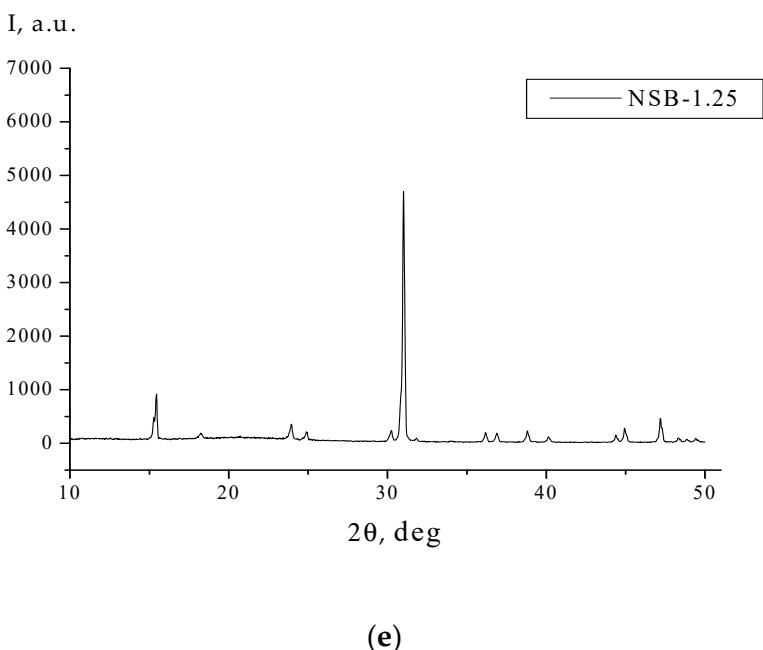

**(e)**

**Figure 12.** Theoretical diffraction patterns for the (**a**) huntite structure (space group *R*32), (**b**) superstructure of the huntite structure (space group *P*321), (**c**) structure with the space group *P*3$_1$21; Experimental diffraction patterns of powdered single crystals (space group *P*321): (**c**) Pr$_{1.1}$Sc$_{2.9}$(BO$_3$)$_4$ (PSB–1.1), (**d**) NdSc$_3$(BO$_3$)$_4$ (NSB–1.0), (**e**) Nd$_{1.25}$Sc$_{2.75}$(BO$_3$)$_4$ (NSB–1.25).

Figure 12d–f show the experimental diffraction patterns of powdered PSB–1.1, NSB–1.0, and NSB–1.25 crystals with the space group *P*321, from which their similarity (with a certain redistribution of the intensities of a series of reflections) and an analogy with the theoretical diffraction patterns is also seen. In order to establish the space group, *R*32 and *P*321, in which the rare-earth scandium orthoborates crystallize, it is necessary to perform, for example, a single-crystal XRD experiment with an analysis of intensities of reflections, as was performed for the PSB and NSB crystals [82].

### 3.2. Solid Solutions of Rare-Earth Scandium Borates

The symmetry of solid solutions in the *Ln*Sc$_3$(BO$_3$)$_4$ – *Ln'*Sc$_3$(BO$_3$)$_4$, *Ln*Sc$_3$(BO$_3$)$_4$ – *Ln'*Sc$_3$(BO$_3$)$_4$ – ScSc$_3$(BO$_3$)$_4$ (space group *R*$\bar{3}$*c*), *Ln*Sc$_3$(BO$_3$)$_4$ – *Ln'*Sc$_3$(BO$_3$)$_4$ – *Ln''*Sc$_3$(BO$_3$)$_4$ systems is determined by the space group and the ratio of the system components. In the LaSc$_3$(BO$_3$)$_4$-CeSc$_3$(BO$_3$)$_4$ system, as well as in the LaSc$_3$(BO$_3$)$_4$ – PrSc$_3$(BO$_3$)$_4$ and CeSc$_3$(BO$_3$)$_4$ – PrSc$_3$(BO$_3$)$_4$ systems, provided that the PrSc$_3$(BO$_3$)$_4$ crystallizes in the space group *C*2/*c*, a formation of a continuous series of solid solutions with the space group *C*2/*c* should be expected. Limited solid solutions, which can be based on the compounds that actually exist, are formed in systems where the final members of solid solution belong to different symmetries (monoclinic and trigonal) (Figure 11). This obvious crystallochemical conclusion, based on the theory of isomorphic mixing, is confirmed experimentally (Table 2).

**Table 2.** Summary structural data on solid solutions of rare-earth scandium borates.

| Initial Composition | Synthesis Method [1] | Space Group | Refined Composition (Method) [2] | Unit Cell Parameters, $a$, $c$, Å | Reference |
|---|---|---|---|---|---|
| | | *$LaSc_3(BO_3)_4$-$NdSc_3(BO_3)_4$ System* | | | |
| $(La_{1-x}Nd_x)Sc_3(BO_3)_4$ (x = 1.0) | SSR and Flux | $R32$ | | | [25] |
| $(La_{1-x}Nd_x)Sc_3(BO_3)_4$ (x ≤ 0.5) | SSR and Flux | $R32$ or $C2/c$ (depending on the crystallized temperature) | | | [25] |
| $(La_{1-x}Nd_x)Sc_3(BO_3)_4$ (x = 0.0–0.3) | Czochralski | $C2/c$ | | | [35,87] |
| $(La_{1-x}Nd_x)Sc_3(BO_3)_4$ (x ≥ 0.5) | Czochralski | $R32$ | | | [100] |
| | | *$LaSc_3(BO_3)_4$-«$GdSc_3(BO_3)_4$»System* | | | |
| $La_xGd_{1-x}Sc_3(BO_3)_4$ (0.20 ≤ x ≤ 0.80) | SSR | $R32$ | | | [101,102] |
| $La_{1-x}Gd_xSc_3(BO_3)_4$ (0.3 ≤ x ≤ 0.7) | Flux | $R32$ | | | [88] |
| $La_{1-x}Gd_xSc_3(BO_3)_4$ (x ≥ 0.3) | SSR | $R32$ | | | [103,104] |
| $La_{0.6}Gd_{0.4}Sc_3(BO_3)_4$, | TSSG | $R32$ | $La_{0.77}Gd_{0.22}Sc_{3.01}(BO_3)_4$ | | [101] |
| $La_{0.4}Gd_{0.6}Sc_3(BO_3)_4$, | TSSG | $R32$ | $La_{0.64}Gd_{0.35}Sc_{3.01}(BO_3)_4$ (ICP) | | [101] |
| $La_{0.2}Gd_{0.8}Sc_3(BO_3)_4$ | TSSG | $R32$ | $La_{0.46}Gd_{0.56}Sc_{2.98}(BO_3)_4$ (ICP) | | [101] |
| $La_xGd_{1-x}Sc_3(BO_3)_4$ | TSSG | $R32$ | $La_{0.78}Gd_{0.22}Sc_3(BO_3)_4$ (ICP) | 9.7933 7.9540 | [101] |
| $La_{0.678}Gd_{0.572}Sc_{2.75}(BO_3)_4$ | Czochralski | $R32$ | $La_{0.64}Gd_{0.41}Sc_{2.95}(BO_3)_4$ (ICP) | 9.794(4) 7.961(6) | [103] |
| $La_{0.75}Gd_{0.5}Sc_{2.75}(BO_3)_4$ | Czochralski | $R32$ | | 9.791 7.952 | [104] |
| | | *$LaSc_3(BO_3)_4$-«$YSc_3(BO_3)_4$»System* | | | |
| $Y_xLa_ySc_z(BO_3)_4$ (0.29 < x < 0.67, 0.67 < y < 0.82, 2.64 < z < 3.00) | SSR | $R32$ | | | [31,102,105] |
| $La_{0.77}Y_{0.28}Sc_{2.95}(BO_3)_4$, $La_{0.76}Y_{0.32}Sc_{2.92}(BO_3)_4$, $La_{0.80}Y_{0.38}Sc_{2.82}(BO_3)_4$, $La_{0.73}Y_{0.42}Sc_{2.85}(BO_3)_4$, $La_{0.75}Y_{0.47}Sc_{2.78}(BO_3)_4$ | TSSG | $R32$ | | | [31,102,105] |
| $Y_2O_3:La_2O_3:Sc_2O_3:B_2O_3:Li_2O$ = 0.6:0.35:1.5:7:7.5 | TSSG | $R32$ | $La_{0.72}Y_{0.57}Sc_{2.71}(BO_3)_4$ (ICP) | 9.774(1) 7.944(3) | [31,102,105–107] |
| $Y_xLa_{1-x}Sc_z(BO_3)_4$ | Flux | $R32$ | $La_{0.75}Y_{0.25}Sc_3(BO_3)_4$ (XRPD) | 9.805(3) 7.980(2) | [108] |
| $La_{0.72}Y_{0.57}Sc_{2.71}(BO_3)_4$ | TSSG | $R32$ | $La_{0.826}Y_{0.334}Sc_{2.84}(BO_3)_4$ (ICP) | 9.8185 7.9893 | [109] |

**Table 2.** *Cont.*

| Initial Composition | Synthesis Method [1] | Space Group | Refined Composition (Method) [2] | Unit Cell Parameters, *a*, *c*, Å | Reference |
|---|---|---|---|---|---|
| *LaSc$_3$(BO$_3$)$_4$-«ErSc$_3$(BO$_3$)$_4$»System* | | | | | |
| LaEr$_{0.006}$Sc$_{2.994}$(BO$_3$)$_{3.8}$<br>LaEr$_{0.015}$Sc$_{2.985}$(BO$_3$)$_{3.8}$ | Czochralski | C2/*c* | | | [35] |
| *LaSc$_3$(BO$_3$)$_4$-«YbSc$_3$(BO$_3$)$_4$»System* | | | | | |
| LaYb$_{0.15}$Sc$_{2.85}$(BO$_3$)$_{3.8}$,<br>LaYb$_{0.3}$Sc$_{2.7}$(BO$_3$)$_{3.8}$,<br>LaYb$_{0.36}$Sc$_{2.64}$(BO$_3$)$_{3.8}$ | Czochralski | C2/*c* | | | [35] |
| *LaSc$_3$(BO$_3$)$_4$-«LuSc$_3$(BO$_3$)$_4$»System* | | | | | |
| La$_x$Lu$_y$Sc$_z$(BO$_3$)$_4$ (x + y + z = 4) | TSSG | R32 | (Lu$_{0.05}$La$_{0.95}$)(Lu$_{0.61}$Sc$_{2.39}$)(BO$_3$)$_4$ ≡<br>La$_{0.95}$Lu$_{0.66}$Sc$_{2.39}$(BO$_3$)$_4$ (ICP) | 9.87420(10)<br>8.0696(7) | [110] |
| *LaSc$_3$(BO$_3$)$_4$-«BiSc$_3$(BO$_3$)$_4$»System* | | | | | |
| La$_x$Bi$_y$Sc$_z$(BO$_3$)$_4$<br>(0.27 < x < 0.52<br>0.67 < y < 0.82<br>2.74 < z < 2.95) | SSR | R32 | | | [111] |
| La$_2$O$_3$ : Sc$_2$O$_3$ : Bi$_2$O$_3$ : B$_2$O$_3$ =1 : 1.5 :<br>13 : 13 | Flux | R32 | La$_{0.82}$Bi$_{0.27}$Sc$_{2.91}$(BO$_3$)$_4$ (ICP) | 9.828(4)<br>7.989(7) | [111] |
| La$_2$O$_3$ : Sc$_2$O$_3$ : Bi$_2$O$_3$ : B$_2$O$_3$ =<br>1 : 1.5 : 13 : 13 | Flux | R32 | La$_{0.91}$Bi$_{0.21}$Sc$_{2.88}$(BO$_3$)$_4$ (EDS) | 9.8370(14)<br>7.9860(14) | [112] |
| *CeSc$_3$(BO$_3$)$_4$-NdSc$_3$(BO$_3$)$_4$System* | | | | | |
| Ce$_{0.53}$Nd$_{0.64}$Sc$_{2.83}$(BO$_3$)$_4$ | Czochralski | P321 | (Ce$_{0.5}$Nd$_{0.5(1)}$)Sc$_3$(BO$_3$)$_4$ (XRD; P321 [3]) | | [35] |
| *CeSc$_3$(BO$_3$)$_4$-«GdSc$_3$(BO$_3$)$_4$»System* | | | | | |
| (Ce$_{0.7}$Gd$_{0.3}$)Sc$_3$(BO$_3$)$_4$ | Czochralski | P321 | | | present work |
| (Ce$_{0.8}$Gd$_{0.2}$)Sc$_3$(BO$_3$)$_4$ | Czochralski | P321 | (Ce$_{0.485(3)}$Gd$_{0.009}$Sc$_{0.006}$)(Ce$_{0.465(6)}$Gd$_{0.017}$Sc$_{0.018}$)Sc$_3$(BO$_3$)$_4$<br>(XRD; P321<br>[3])(Ce$_{0.803(2)}$Gd$_{0.091(4)}$Sc$_{0.106(4)}$)Sc$_3$(BO$_3$)$_4$<br>(XRD; R32 [3]) | 9.7812(10)<br>7.9480(12) | present work |
| Ce$_{0.9}$Gd$_{0.35}$Sc$_{2.75}$(BO$_3$)$_4$ | Czochralski | P321 | (Ce$_{0.949(1)}$Gd$_{0.024(1)}$Sc$_{0.027(1)}$)Sc$_3$(BO$_3$)$_4$<br>(XRD; R32 [3]) | 9.7776(42)<br>7.9436(21) | present work |
| *CeSc$_3$(BO$_3$)$_4$-«YSc$_3$(BO$_3$)$_4$» System* | | | | | |
| Ce$_{1.25}$Y$_{0.3}$Sc$_{2.45}$(BO$_3$)$_4$ | Czochralski | P321 | (Ce$_{0.78}$Y$_{0.22(2)}$)Sc$_3$(BO$_3$)$_4$ (XRPD; R32 [3]) | | [35] |
| Ce$_{1.25}$Y$_{0.3}$Sc$_{2.45}$(BO$_3$)$_4$ | Czochralski | P321 | (Ce$_{0.995(1)}$Y$_{0.002(1)}$Sc$_{0.003(1)}$)Sc$_3$(BO$_3$)$_4$<br>(XRD; R32 [3]) | 9.7553(25)<br>7.9680(12) | present work |

**Table 2.** *Cont.*

| Initial Composition | Synthesis Method [1] | Space Group | Refined Composition (Method) [2] | Unit Cell Parameters, $a$, $c$, Å | Reference |
|---|---|---|---|---|---|
| *CeSc$_3$(BO$_3$)$_4$-«LuSc$_3$(BO$_3$)$_4$» System* | | | | | |
| Ce$_{1.25}$Lu$_{0.3}$Sc$_{2.45}$(BO$_3$)$_4$ | Czochralski | *P*321 | Ce(Lu$_{0.17(1)}$Sc$_{2.83}$)(BO$_3$)$_4$ (XRPD; *R*32 [3]) | | [35] |
| Ce$_{1.25}$Lu$_{0.3}$Sc$_{2.45}$(BO$_3$)$_4$ | Czochralski | *P*321 | Ce(Sc$_{2.910(30)}$Lu$_{0.090}$)(BO$_3$)$_4$ (XRD; *R*32 [3]) | 9.8085(21) 7.9829(10) | present work |
| *PrSc$_3$(BO$_3$)$_4$-NdSc$_3$(BO$_3$)$_4$»System* | | | | | |
| Pr$_{0.99}$Nd$_{0.11}$Sc$_{2.9}$(BO$_3$)$_4$ | Czochralski | *P*321 | | | [83] |
| *PrSc$_3$(BO$_3$)$_4$-«YSc$_3$(BO$_3$)$_4$»System* | | | | | |
| Nd$_2$O$_3$ : Y$_2$O$_3$ : Sc$_2$O$_3$ : HBO$_2$ : Li$_2$(CO$_3$) : LiF = 0.25 : 0.25 : 0.8 : 2.75 : 0.177 : 0.246 | TSSG | *R*32 | Pr$_{0.94}$Y$_{0.09}$Sc$_{2.97}$(BO$_3$)$_4$ (SEM/EDX) (periphery part) Pr$_{0.93}$Y$_{0.10}$Sc$_{2.96}$(BO$_3$)$_4$ (SEM/EDX) (central part) | 9.8256(5) 7.9038(4) 9.8179(6) 7.9029(1) | [113] |
| *NdSc$_3$(BO$_3$)$_4$-«GdSc$_3$(BO$_3$)$_4$»System* | | | | | |
| Nd$_{1.125}$Gd$_{0.125}$Sc$_{2.75}$(BO$_3$)$_4$ | Czochralski | *P*321 | (Nd$_{0.8}$Gd$_{0.2(1)}$)Sc$_3$(BO$_3$)$_4$ (XRD; *P*321 [3]) | | [35] |
| *NdSc$_3$(BO$_3$)$_4$-«YSc$_3$(BO$_3$)$_4$»System* | | | | | |
| Pr$_2$O$_3$ : Y$_2$O$_3$ : Sc$_2$O$_3$ : HBO$_2$ : Li$_2$(CO$_3$) : LiF = 0.125 : 0.125 : 0.5 : 1.94 : 0.221 : 0.205 | TSSG | *R*32 | Nd$_{0.86}$Y$_{0.21}$Sc$_{2.93}$(BO$_3$)$_4$ (SEM/EDX) (periphery part) Nd$_{0.87}$Y$_{0.18}$Sc$_{2.95}$(BO$_3$)$_4$ (SEM/EDX) (part under the seed) | 9.7588(5) 7.9186(9) 9.7631(3)7.9210(7) | [113] |
| *CeSc$_3$(BO$_3$)$_4$-NdSc$_3$(BO$_3$)$_4$-«GdSc$_3$(BO$_3$)$_4$»System* | | | | | |
| (Ce$_{0.65}$Nd$_{0.25}$Gd$_{0.10}$)Sc$_3$(BO$_3$)$_4$ | Czochralski | *P*321 | (Ce$_{0.41(4)}$Nd$_{0.46}$Gd$_{0.13}$)Sc$_3$(BO$_3$)$_4$ (XRD; *R*32 [3]) | | [35] |
| Ce$_{0.76}$Nd$_{0.30}$Gd$_{0.14}$Sc$_{2.8}$(BO$_3$)$_4$ | Czochralski | *P*321 | (Ce$_{0.57}$Nd$_{0.25}$Gd$_{0.18}$)Sc$_3$(BO$_3$)$_4$ (XRD; *P*321 [3]) | | [35] |

**Table 2.** *Cont.*

| Initial Composition | Synthesis Method [1] | Space Group | Refined Composition (Method) [2] | Unit Cell Parameters, *a, c*, Å | Reference |
|---|---|---|---|---|---|
| | | *CeSc₃(BO₃)₄-NdSc₃(BO₃)₄-«LuSc₃(BO₃)₄»System* | | | |
| $Ce_{1.2}Nd_{0.05}Lu_{0.3}Sc_{2.45}(BO_3)_4$ | Czochralski | $P321$ | | 21 reflections (space group *R*32): <br> 9.776(6) <br> 7.937(5) <br> 18 reflections (space group *A*2): <br> $a' = 7.895$, <br> $b' = 9.749$, <br> $c' = 16.817$, <br> $\alpha = 90.23°$, <br> $\beta = 89.95°$, <br> $\gamma = 90.18°$ | [35] |
| | | *LaSc₃(BO₃)₄-«ErSc₃(BO₃)₄»-«YbSc₃(BO₃)₄»System* | | | |
| $LaYb_{0.15}Er_{0.015}Sc_{2.835}(BO_3)_{3.8}$, <br> $LaYb_{0.3}Er_{0.015}Sc_{2.685}(BO_3)_{3.8}$, <br> $LaYb_{0.3}Er_{0.003}Sc_{2.67}(BO_3)_{3.8}$, <br> $LaYb_{0.36}Er_{0.015}Sc_{2.625}(BO_3)_{3.8}$ | Czochralski | $C2/c$ | | | [35] |

[1] SSR—solid-state reaction; TSSG—top-seeded solution growth.[2] ICP—inductively coupled plasma elemental analysis; XRPD—X-ray powder diffraction; EDS—energy dispersive spectra standardless analysis; XRD—X-ray (single-crystal) diffraction; SEM/EDX—scanning electron microscopy with energy dispersive X-ray spectroscopy. [3] A structure and composition were refined in the space group specified.

### 3.2.1. LaSc$_3$(BO$_3$)$_4$-NdSc$_3$(BO$_3$)$_4$ System

The XRD study of the (La$_{1-x}$Nd$_x$)Sc$_3$(BO$_3$)$_4$ (LNSB) solid solutions ($x$ = 0.0–0.3) grown by the Czochralski method showed their crystallization in the space group $C2/c$ [35,87]. The non-centrosymmetric trigonal phase $R32$ was discovered only in some crystals with high Nd$^{3+}$ concentration ($x \geq 0.5$) [100]. A trigonal symmetry was found only for NSB crystal (Ir seed). For LSB (Ir seed) as well as (Nd$_{0.5}$La$_{0.5}$)Sc$_3$(BO$_3$)$_4$ and (Nd$_{0.1}$La$_{0.9}$)Sc$_3$(BO$_3$)$_4$ with the LSB seed, a monoclinic symmetry was established. It confirms a role of a seed in symmetry of grown crystals.

Symmetry of LNSB solid solutions ($x$ = 0 to 1) obtained by the solid-state reaction and the flux method, was demonstrated by DTA, X-ray diffraction, second-harmonic generation, and fluorescence lifetime measurements [25]: for $x$ = 1.0, i.e., for NdSc$_3$(BO$_3$)$_4$, only trigonal phase existed, and for $x \leq 0.5$, in the LNSB sample, two phases, with the space group $R32$ or $C2/c$, were obtained depended on the crystallized temperature (according to powder XRD, the phase with the space group $R32$ could be formed below 1100 °C). For the LSB ($x$ = 0), the pure trigonal phase was difficult to obtain from the solid-state reaction. For example, in the LSB sample sintered at 1000 °C for 2 h, the huntite-family monoclinic phase and LaBO$_3$ were found.

### 3.2.2. LaSc$_3$(BO$_3$)$_4$ -«GdSc$_3$(BO$_3$)$_4$» System

Polycrystalline La$_x$Gd$_{1-x}$Sc$_3$(BO$_3$)$_4$ (LGSB) solid solutions were obtained by the conventional solid-state reaction [101–104], and single-crystal ones were grown by the high-temperature TSSG method using the Li$_6$B$_4$O$_9$–LiF as a flux [101,102] and Czochralski method [103,104] (Table 2). A crystallization of polycrystalline samples in the space group $R32$ for $0.20 \leq x \leq 0.80$ was established [101,102]. According to the XRD analysis, a single crystal grown by the high-temperature TSSG method crystallizes in the space group $R32$ and have the composition La$_{0.78}$Gd$_{0.22}$Sc$_3$(BO$_3$)$_4$, refined using a full-matrix least-squares refinement on $F^2$ (SHXLXL-97) using the occupancy factor for the La site fixed to a value that was determined from the ICP elemental analysis [101]. The composition of the single crystal grown by the Czochralski method from the charge La$_{0.678}$Gd$_{0.572}$Sc$_{2.75}$(BO$_3$)$_4$ is determined by the inductively coupled plasma atomic emission spectrophotometry (ICP-AES) method to be La$_{0.64}$Gd$_{0.41}$Sc$_{2.95}$(BO$_3$)$_4$ (space group $R32$) [103]. In addition, the LGSB single crystals were obtained by the Czochralski method from the charge with initial composition La$_{0.75}$Gd$_{0.5}$Sc$_{2.75}$(BO$_3$)$_4$ without refinement of crystal real composition [104].

According to [101], as a result of investigation of polycrystalline LGSB samples, it was revealed that the stoichiometry of the single trigonal phase followed the relationship of Sc/(Gd + La) = 3 and the deviations from these values afford additional peaks of impurities in the X-ray powder diffraction patterns. Therefore, Gd occupies the La site rather than Sc site [100]. It is confirmed by the Gheorghe et al. [103]: according to the ICP-AES method, the stoichiometry of Sc is close to 3. According to [103,104], for polycrystalline La$_{1-x}$Gd$_x$Sc$_3$(BO$_3$)$_4$ solid solutions, it was established that the structural changes from monoclinic (space group $C2/c$) to trigonal (space group $R32$) is complete for a Gd content larger than 0.3. The XRD investigation of a series of single crystal LGSB solid solutions, grown in 3:1 wt/wt LiBO$_2$/La$_{1-x}$Gd$_x$Sc$_3$(BO$_3$)$_4$ solvents or a 20 wt% CaF$_2$ flux, showed that the $R32$ structural integrity is maintained for $0.3 \leq x \leq 0.7$ [88].

### 3.2.3. LaSc$_3$(BO$_3$)$_4$ -«YSc$_3$(BO$_3$)$_4$» System

Polycrystalline and single-crystal La$_x$Y$_{1-x}$Sc$_3$(BO$_3$)$_4$ (LYSB) solid solutions were obtained by the solid-state reaction [31,102,105] and the high-temperature TSSG method using a lithium-borate flux [31,102,105–108], respectively (Table 2). An investigation of phase equilibria in the LaBO$_3$-ScBO$_3$-YBO$_3$ system [31,102,105] allowed to establish the limits of existence of huntite-type trigonal (space group $R32$) Y$_x$La$_y$Sc$_z$(BO$_3$)$_4$ solid solutions: $0.29 < x < 0.67$, $0.67 < y < 0.82$, and $2.64 < z < 3.00$. Selected initial compositions of samples within the trigonal-huntite region of the phase diagram are given in [31,102,105] (Table 2). The space group $R32$ has been determined for these

samples by the powder XRD, and then was proven on the basis of the X-ray single-crystal refinement of structure of $La_{0.72}Y_{0.57}Sc_{2.71}(BO_3)_4$. Occupancy factors for the La and Sc sites were fixed to the values that were determined by ICP elemental analysis, as deviations from these occupancies afforded increased residuals in the least-squares refinements. The solid solution with the trigonal symmetry, grown by the TSSG method, had an average composition $La_{0.826}Y_{0.334}Sc_{2.84}(BO_3)_4$, determined by the ICP-AES [109].

According to the refinement of atomic coordinates and displacement parameters of LYSB structure of single crystal using a full-matrix least squares refinement on $F^2$ (SHELXL-97) [31], its composition can be written as $La_{0.72}Y_{0.57}Sc_{2.71}(BO_3)_4$ (space group $R32$), occupancy factors for the La, Y and Sc sites being fixed to the values determined by ICP-AES elemental analysis. Based on the comparison of cell volumes for the compositions $La_{0.77}Y_{0.28}Sc_{2.95}(BO_3)_4$ and $Y_{0.47}La_{0.75}Sc_{2.78}(BO_3)_4$ (La, Y, and Sc occupancies were determined by ICP-AES elemental analysis), in which the La concentrations are similar, the higher Y concentration of $La_{0.75}Y_{0.47}Sc_{2.78}(BO_3)_4$ ($V$ = 670.5(1) Å$^3$) relative to that of $La_{0.77}Y_{0.28}Sc_{2.95}(BO_3)_4$ ($V$ = 666.35(9) Å$^3$) leads to a 0.6% increase in cell volume, which can only occur if the Y atom substitutes the small Sc site [31]. The structure of $La_{0.75}Y_{0.25}Sc_3(BO_3)_4$ single crystal (space group $R32$) grown by the flux method was determined [108]. After full-matrix refinement with isotropic displacement coefficients on each atom, the occupancies of the La and Sc sites were refined. No significant change in the Sc occupancy factor was observed, so it was subsequently fixed to unity. Occupancy of the La site was significantly reduced, indicating occupation of Y on the La site. On the basis of the systematic condition $-h + k + l = 3n$ for the *hkil*, the statistical analysis of the intensity distribution, packing considerations, and the successful solution and refinement of the structure, the crystal was found to form in the non-centrosymmetric space group $R32$.

### 3.2.4. $LaSc_3(BO_3)_4$ -«$ErSc_3(BO_3)_4$» System

Durmanov et al. [35] selected the optimal compositions of charge with a $B_2O_3$ deficiency, which prevents its deposition on the surface of the growing crystal, since liquid $B_2O_3$ drains into the high temperature region and dissolves the growing crystal. As a result, optically qualitative activated $LaEr_{0.006}Sc_{2.994}(BO_3)_{3.8}$ and $LaEr_{0.015}Sc_{2.985}(BO_3)_{3.8}$ crystals with the space group $C2/c$ were obtained by the Czochralski method. Based on the crystallochemical similarity of Er with the Lu and Tb atoms, which, according to the XRD investigations, enter the Sc site [87], Durmanov et al. [35] suggested that the $Er^{3+}$ ions also replace the $Sc^{3+}$ ones. An increase in the concentration of $Er^{3+}$ ions in the initial melt above 12 at % can lead to cracking of crystals and a change in their symmetry.

### 3.2.5. $LaSc_3(BO_3)_4$ -«$YbSc_3(BO_3)_4$» System

The optimal compositions of the charge with the lack of $B_2O_3$ to grow optically-qualitative crystals with the space group $C2/c$ by the Czochralski method are $LaYb_{0.15}Sc_{2.85}(BO_3)_{3.8}$, $LaYb_{0.3}Sc_{2.7}(BO_3)_{3.8}$, $LaYb_{0.36}Sc_{2.64}(BO_3)_{3.8}$ [35]. Durmanov et al. [35] suggested that the $Yb^{3+}$ ions enter the Sc site based on the crystallochemical similarity of Yb with the Lu and Tb, which, according to the XRD investigations, occupy the Sc sites [87]. The maximum allowable concentration of $Yb^{3+}$ ions to grow laser crystals is 10 at % ($1.3 \times 10^{21}$ cm$^{-3}$); higher concentration cannot be achieved using the growth technology described in [35].

### 3.2.6. $LaSc_3(BO_3)_4$ -«$LuSc_3(BO_3)_4$» System

Polycrystalline and single-crystal solid solutions with the initial composition $La_xLu_ySc_z(BO_3)_4$ ($x + y + z = 4$) were obtained by the solid-state reaction and the high-temperature TSSG method with a $Li_6B_4O_9$ flux [110], respectively (Table 2). According to single-crystal XRD measurements, the solid solution having the composition $La_{0.95}Lu_{0.66}Sc_{2.39}(BO_3)_4$, determined from the ICP elemental analysis, has been found to crystallize in the space group $R32$. Li et al. [110] suggested that the Lu atoms replace not only the La atoms in the trigonal prisms but also the Sc atoms in the octahedra; the formula for demonstrating Lu-atom substitution can be written as $(Lu_{0.05}La_{0.95})(Lu_{0.61}Sc_{2.39})(BO_3)_4$.

### 3.2.7. LaSc$_3$(BO$_3$)$_4$ -«BiSc$_3$(BO$_3$)$_4$» System

Solid solutions with the general composition La$_x$Bi$_y$Sc$_z$(BO$_3$)$_4$ were obtained in the form of polycrystals and single crystals using the solid-state reaction [111] and the spontaneous crystallization method using the Bi$_2$O$_3$–B$_2$O$_3$ as a flux [111,112], respectively (Table 2). Xu et al. [111] established that the single trigonal phase covers the composition range of $0.27 < x < 0.52$, $0.67 < y < 0.82$, and $2.74 < z < 2.95$. According to Xu et. al. [111], the Bi atoms replace not only the La atoms in the trigonal prisms but also the Sc atoms in the octahedra, and, hence, the general composition can be written as (La$_{1-x}$Bi$_x$)(Bi$_y$Sc$_{3-y}$)(BO$_3$)$_4$. The unit-cell volume ($V$) varies with the sizes of both the trigonal prism and the octahedron, and it can be expressed in the equation by using atomic radii ($R$) and appropriately weighted occupancies for the Bi, La, and Sc atoms. Based on the equation given in [111], an approximately linear relationship between the effective concentration ($Ceff$) and $V$ was obtained, which indicates that the Bi atoms do indeed appear to be distributed across both sites. For the single crystal with the composition La$_{0.82}$Bi$_{0.27}$Sc$_{2.91}$(BO$_3$)$_4$, determined from the ICP-AES elemental analysis, the space group $R32$ was found using the XRD analysis [111] (Table 2).

Single-crystal La$_{0.91}$Bi$_{0.21}$Sc$_{2.88}$(BO$_3$)$_4$ solid solution, obtained by the spontaneous crystallization, the composition of which was checked with the energy dispersive spectra standardless analysis, has the trigonal structure with the space group R32, atomic sites and isotropic displacement factors being refined without any refinement of site occupancies [112] (Table 2).

### 3.2.8. CeSc$_3$(BO$_3$)$_4$-NdSc$_3$(BO$_3$)$_4$ System

Single-crystal solid solution with the composition of the initial charge (Ce$_{0.53}$Nd$_{0.64}$Sc$_{2.83}$)(BO$_3$)$_4$, grown by the Czochralski method, crystallizes in the space group $P321$ [35]. A comparison of the unit cell parameters, atomic coordinates, and interatomic distances of this phase with those for the NdSc$_3$(BO$_3$)$_4$ allowed to receive the composition (Ce$_{0.5}$Nd$_{0.5(1)}$)Sc$_3$(BO$_3$)$_4$. The structure disordering of (Ce$_{0.5}$Nd$_{0.5(1)}$)Sc$_3$(BO$_3$)$_4$ is more pronounced compared with the NdSc$_3$(BO$_3$)$_4$ [35].

### 3.2.9. CeSc$_3$(BO$_3$)$_4$-«GdSc$_3$(BO$_3$)$_4$» System

Single-crystal solid solutions obtained by the Czochralski method from the charges having compositions (Ce$_{0.8}$Gd$_{0.2}$)Sc$_3$(BO$_3$)$_4$, (Ce$_{0.7}$Gd$_{0.3}$)Sc$_3$(BO$_3$)$_4$, and Ce$_{0.9}$Gd$_{0.35}$Sc$_{2.75}$(BO$_3$)$_4$ crystallize in the space group $P321$ (Table 2). According to the XRD analysis of microcrystals, 67% of diffraction reflections do not obey the extinction laws of the space group $R32$, but they are indexed in the space group $P321$, i.e., in the superstructure relative to the huntite structure. A refinement of the crystal structure of solid solutions with the nominal compositions (Ce$_{0.80}$Gd$_{0.20}$)Sc$_3$(BO$_3$)$_4$ and Ce$_{0.9}$Gd$_{0.35}$Sc$_{2.75}$(BO$_3$)$_4$ based on the strongest reflections (33%) in the subcell with the space group $R32$, i.e., in the huntite structure (atomic coordinates, atomic displacement parameters, occupancies of all crystallographic sites except for B and O ones), showed that their compositions can be described as (Ce$_{0.803(2)}$Gd$_{0.091(4)}$Sc$_{0.106(4)}$)Sc$_3$(BO$_3$)$_4$ ($R$ = 3.04 %) and (Ce$_{0.949(1)}$Gd$_{0.024(1)}$Sc$_{0.027(1)}$)Sc$_3$(BO$_3$)$_4$ ($R$ = 1.93 %), respectively. The refined crystal compositions correlate with the unit cell parameters ($r_{Ce} > r_{Gd} > r_{Sc}$), which are indicators of a composition. A refinement of the structure of the microcrystal with the nominal composition (Ce$_{0.80}$Gd$_{0.20}$)Sc$_3$(BO$_3$)$_4$ in the space group $P321$ (i.e., taking into account all diffraction reflections, both strong and weak, but with the $I \geq 3\sigma(I)$; $R$ = 3.13%) allowed writing the composition as (Ce$_{0.485(3)}$Gd$_{0.009}$Sc$_{0.006}$)(Ce$_{0.465(6)}$Gd$_{0.017}$Sc$_{0.018}$)Sc$_3$(BO$_3$)$_4$ or (Ce$_{0.950(3)}$Gd$_{0.026}$Sc$_{0.024}$)Sc$_3$(BO$_3$)$_4$ (Tables 3 and 4).

**Table 3.** Coordinates of atoms, atom displacement parameters ($B_{eq} \times 10^2$, Å$^2$), and site occupancies ($p$) in the structure of $(Ce_{0.80}Gd_{0.20})Sc_3(BO_3)_4$ solid solution in the space group $R32$ and $P321$ according to the XRD data (Ag$K\alpha$).

| Parameter | Space Group $R32$ | Space Group $P321$ | Parameter | Space Group $R32$ | Space Group $P321$ | Parameter | Space Group $R32$ | Space Group $P321$ |
|---|---|---|---|---|---|---|---|---|
| **Z** | 3 | | **μ, mm$^{-1}$** | 3.90 | 3.87 | **θ$_{max}$, deg** | 25.865 | |
| **a, Å** | 9.781(1) | | **Refl.: read/unique** | 1765/591 | 1765/1765 | **wR$_2$** | 0.0500 | 0.0753 |
| **c, Å** | 7.948(1) | | **(I > 2σ(I))** | | | **R$_1$ (I > 2σ(I))** | 0.0304 | 0.0313 |
| **V, Å$^3$** | 658.53 | | **No. of parameters** | 37 | 90 | **S** | 0.857 | 0.978 |
| *Site* | Ce1/Gd1/Sc | Ce1/Gd1/Sc | *Site* | | B2 | *Site* | | O3 |
| *x* | 0 | 0 | *x* | | 0.3282(4) | *x* | | 0.3114(3) |
| *y* | 0 | 0 | *y* | | 0.1146(5) | *y* | | 0.2479(4) |
| *z* | 0 | 0 | *z* | | 0.1705(5) | *z* | | 0.1754(3) |
| *B$_{eq}$* | 1.06(1) | 1.11(1) | *B$_{eq}$* | | 1.28(6) | *B$_{eq}$* | | 1.96(8) |
| *p*(Ce1/Gd1/Sc1) | 0.1338(3)/0.0152(7)/0.0177(7) | 0.1617(9)/0.0030(9)/0.0020(9) | *p* | | 1.0 | *p* | | 1.0 |
| *Site* | | Ce2/Gd2/Sc | *Site* | B2 | B3 | *Site* | | O4 |
| *x* | | 1/3 | *x* | 0 | 0 | *x* | | 0.4755(4) |
| *y* | | 2/3 | *y* | 0 | 0 | *y* | | 0.1243(3) |
| *z* | | 0.66638(4) | *z* | 0.5 | 0.5 | *z* | | 0.1562(3) |
| *B$_{eq}$* | | 1.04(1) | *B$_{eq}$* | 1.3(1) | 1.4(2) | *B$_{eq}$* | | 1.56(5) |
| *p*(Ce2/Gd2/Sc2) | | 0.155(2)/0.006(2)/0.006(2) | *p* | 0.16667 | 0.16667 | *p* | | 1.0 |
| *Site* | Sc1 | Sc1 | *Site* | | B4 | *Site* | O2 | O5 |
| *x* | 0.2141(1) | 0.55053(9) | *x* | | 2/3 | *x* | 0.58777 | 0.5989(5) |
| *y* | 1/3 | 0 | *y* | | 1/3 | *y* | 0 | 0 |
| *z* | 1/3 | 0 | *z* | | 0.8473(6) | *z* | 0.5 | 0.5 |
| *B$_{eq}$* | 0.92(3) | 0.92(2) | *B$_{eq}$* | | 0.82(9) | *B$_{eq}$* | 2.4(1) | 2.02(8) |
| *p* | 0.5 | 0.5 | *p* | | 0.3333 | *p* | 0.5 | 0.5 |
| *Site* | | Sc2 | *Site* | | O1 | *Site* | O3 | O6 |
| *x* | | 0.21195(9) | *x* | | 0.4560(3) | *x* | −0.1399(5) | 0.1405(4) |
| *y* | | 0.33235(6) | *y* | | 0.1408(3) | *y* | 0 | 0.1405(4) |
| *z* | | 0.32954(6) | *z* | | 0.5228(3) | *z* | 0.5 | 0.5 |
| *B$_{eq}$* | | 0.86(2) | *B$_{eq}$* | | 1.44(5) | *B$_{eq}$* | 1.54(8) | 1.22(4) |
| *p* | | 1.0 | *p* | | 1.0 | *p* | 0.5 | 0.5 |
| *Site* | B1 | B1 | *Site* | O1 | O2 | *Site* | | O7 |
| *x* | 0.4516(6) | 0.4536(5) | *x* | 0.0193(4) | 0.1930(4) | *x* | | 0.6745(3) |
| *y* | 0 | 0 | *y* | 0.2116(4) | −0.0209(3) | *y* | | 0.1983(3) |
| *z* | 0.5 | 0.5 | *z* | 0.1813(4) | 0.1769(3) | *z* | | −0.1536(2) |
| *B$_{eq}$* | 1.30(8) | 1.05(8) | *B$_{eq}$* | 1.58(6) | 1.51(5) | *B$_{eq}$* | | 1.22(4) |
| *p* | 0.16667 | 0.5 | *p* | 1.0 | 1.0 | *p* | | 1.0 |

**Table 4.** Main interatomic distances in the structure of of $(Ce_{0.80}Gd_{0.20})Sc_3(BO_3)_4$ solid solution in the space group *R*32 and *P*321 according to the XRD data (Ag*Kα*).

| Parameter | Space Group *R*32 | Parameter | Space Group *P*321 |
|---|---|---|---|
| Ce1/Gd1/Sc | | Ce1/Gd1/Sc | |
| − 6 × O1 | 2.450(3) | − 6 × O2 | 2.443(3) |
| | | Ce2/Gd2/Sc | |
| | | − 3 × O4 | 2.417(3) |
| | | − 3 × O1 | 2.484(3) |
| | | [Ce2/Gd2/Sc-O]av. | 2.4505 |
| Sc1 | | Sc1 | |
| − 2 × O1 | 2.059(3) | − 2 × O7 | 2.091(3) |
| − 2 × O3 | 2.118(3) | − 2 × O4 | 2.110(3) |
| − 2 × O2 | 2.148(4) | − 2 × O3 | 2.211(3) |
| [Sc1-O]av. | 2.129 | [Sc1-O]av. | 2.134 |
| | | Sc2 | |
| | | − 1 × O2 | 2.035(3) |
| | | − 1 × O1 | 2.041(3) |
| | | − 1 × O5 | 2.113(3) |
| | | − 1 × O3 | 2.124(3) |
| | | − 1 × O6 | 2.129(2) |
| | | − 1 × O7 | 2.149(2) |
| | | [Sc2-O]av. | 2.0985 |
| B1 | | B1 | |
| − 1 × O2 | 1.33(1) | − 2 × O1 | 1.378(4) |
| − 2 × O1 | 1.368(5) | − 1 × O5 | 1.412(8) |
| | 1.355 | [B1-O]av. | 1.389 |
| | | B2 | |
| | | − 1 × O3 | 1.289(6) |
| | | − 1 × O2 | 1.325(5) |
| | | − 1 × O4 | 1.401(5) |
| | | [B2-O]av. | 1.338 |
| B2 | | B3 | |
| − 3 × O3 | 1.369(5) | − 3 × O6 | 1.375(4) |
| | | B4 | |
| | | − 3 × O7 | 1.360(3) |

### 3.2.10. CeSc$_3$(BO$_3$)$_4$-«YSc$_3$(BO$_3$)$_4$» System

A refinement of the crystal structure of the Czochralski-grown powdered crystal with the initial composition Ce$_{1.25}$Y$_{0.3}$Sc$_{2.45}$(BO$_3$)$_4$ by the Rietveld method in the huntite subcell with the space group *R*32 (the proper space group is *P*321: 67% of diffraction reflections do not obey the extinction laws of the space group *R*32) showed its real composition as (Ce$_{0.78}$Y$_{0.22(2)}$)Sc$_3$(BO$_3$)$_4$ [35]. A decrease in the Ce site occupancy was observed (the form factor or atomic factor is proportional to the atomic number), which indicates a partial replacement of the Ce atoms by the Y ones, and the presence of Sc atoms in this site has not been considered. The XRD analysis of the cation sites, except for the B site, in the structure of micropart of the same crystal (space group *R*32) resulted in the composition (Ce$_{0.995(1)}$Y$_{0.002(1)}$Sc$_{0.003(1)}$)Sc$_3$(BO$_3$)$_4$ (*R* = 7.73 %), i.e., with the absence of Y atoms in the Sc site. Different compositions refined for powder and single-crystal samples can be explained by the heterogeneity of bulk crystal composition.

### 3.2.11. CeSc$_3$(BO$_3$)$_4$-«LuSc$_3$(BO$_3$)$_4$» System

A refinement of structure of the powdered sample in the space group *R*32 (the proper space group is *P*321) using the full-profile method showed that the composition of the Czochralski-grown crystal

can be written as $Ce(Lu_{0.17(1)}Sc_{2.83})(BO_3)_4$ with a presence of the Lu ions in the octahedral sites of the structure together with the Sc ones [35]. The XRD analysis of the structure of micropart of the same crystal confirmed the presence of Lu atoms in the octahedral site, but with a lower content: $Ce(Sc_{2.910(30)}Lu_{0.090})(BO_3)_4$ ($R$ = 5.95%).

### 3.2.12. $PrSc_3(BO_3)_4$ -$NdSc_3(BO_3)_4$ System

A solid solution with the charge composition $(Pr_{0.99}Nd_{0.11}Sc_{2.9})(BO_3)_4$ grown by the Czochralski method crystallizes in the space group $P321$ [83].

### 3.2.13. $PrSc_3(BO_3)_4$-«$YSc_3(BO_3)_4$» System

A solid solution with the general composition $Pr_xY_ySc_z(BO_3)_4$ grown by the TSSG method using eutectic $LiBO_2$–$LiF$ as a flux crystallizes in the space group $R32$, as determined by the XRD on the powder sample [113] (Table 2). According to scanning electron microscopy with energy dispersive X-ray spectroscopy (SEM/EDX), a peripheral part of the crystal has the composition $Pr_{0.94}Y_{0.09}Sc_{2.97}(BO_3)_4$, while the composition of the central part of the crystal is $Pr_{0.93}Y_{0.10}Sc_{2.96}(BO_3)_4$ [113]. It should be noted that the unit cell parameters of samples from two parts of the crystal with almost identical compositions are significantly different, although there is a tendency for the parameters to increase with increasing Pr content ($r_{Pr} > r_Y$) (Table 2).

### 3.2.14. $NdSc_3(BO_3)_4$-«$GdSc_3(BO_3)_4$» System

The XRD analysis of the Czochralski-grown crystals with the initial compositions $Nd_{1.125}Gd_{0.125}Sc_{2.75}(BO_3)_4$ (the refined composition $(Nd_{0.8}Gd_{0.2(1)})Sc_3(BO_3)_4$ was evaluated by comparing the structural parameters of this phase with those for the $NdSc_3(BO_3)_4$), $Nd_{1.04}Gd_{0.26}Sc_{2.7}(BO_3)_4$, and $Nd_{0.91}Gd_{0.39}Sc_{2.7}(BO_3)_4$ showed their crystallization in the space group $P321$ [35].

### 3.2.15. $NdSc_3(BO_3)_4$-«$YSc_3(BO_3)_4$» System

Single-crystal solid solution with the charge composition $Nd_xY_ySc_z(BO_3)_4$ was grown by the TSSG method [113]. The space group $R32$ was established by the powder XRD method. According to the SEM/EDX, a periphery part of the crystal has the composition $Nd_{0.86}Y_{0.21}Sc_{2.93}(BO_3)_4$, while the region under the seed is $Nd_{0.87}Y_{0.18}Sc_{2.95}(BO_3)_4$. A situation is similar to that found for solid solutions in the $PrSc_3(BO_3)_4$-«$YSc_3(BO_3)_4$» system: significant differences in the unit cell parameters of samples taken from different parts of the crystal with almost identical compositions (taking into account a measurement error) are found, a noticeable correlation between the unit cell parameters and the Y content ($r_{Nd} > r_Y$) being observed.

### 3.2.16. $CeSc_3(BO_3)_4$-$NdSc_3(BO_3)_4$-«$GdSc_3(BO_3)_4$» System

The XRD study of a solid solution grown by the Czochralski method showed that its structure has the huntite subcell with the space group $R32$ (Table 2). In this case, a limited number of weak diffraction reflections go into a superstructural trigonal cell with parameters doubled with respect to the huntite ones: $A = 2a_{R32}$, $C = 2c_{R32}$ [35]. The single-crystal XRD analysis of a solid solution with the charge composition $Ce_{0.76}Nd_{0.30}Gd_{0.14}Sc_{2.8}(BO_3)_4$ and estimated composition $(Ce_{0.57}Nd_{0.25}Gd_{0.18})Sc_3(BO_3)_4$ revealed a significant number of superstructure reflections with the space group $P321$ [35]. Durmanov et al. [35] believe that the choice of the initial melt composition and growth conditions may rule out a formation of superstructure in the $(Ce,Nd,Gd)Sc_3(BO_3)_4$ solid solution; among these compositions may be found a congruent melting one.

### 3.2.17. $CeSc_3(BO_3)_4$-$NdSc_3(BO_3)_4$-«$LuSc_3(BO_3)_4$» System

A large number of maximally-split reflections was revealed as a result of XRD investigation of a small chip of the Czochralski-grown crystal with the initial composition $Ce_{1.2}Nd_{0.05}Lu_{0.3}Sc_{2.45}(BO_3)_4$.

The unit cell parameters determined by an auto-indexing the most intense 21 reflections correspond to the huntite cell with the parameters $a_{R32}$ = 9.776(6), $c_{R32}$ = 7.937(5) Å [35]. In the interval of interplanar distances $d$ = 2.32–3.14 Å, weak and diffuse reflections, along with the strong ones, were found. The unit cell parameters determined from 18 reflections in the same interval of interplanar distances were found to be $a'$ = 7.895, $b'$ = 9.749, $c'$ = 16.817Å, $\alpha$ = 90.23°, $\beta$ = 89.95°, $\gamma$ = 90.18°. The obtained triclinic unit cell is a pseudo-monoclinic one (space group $A2$) with the parameters correlated with those of the huntite cell: $a'$ = $c_{R32}$, $b'$ = $b_{R32}$, $c'$ = $2a_{R32}\cos30°$, $\alpha \sim 90°$, $\beta \sim 90°$, $\gamma \sim 90°$. However, two weak (the intensities are 44 and 65 times less than the intensity of the strongest reflection in the above-mentioned interval) and diffuse (peak width are 1.56° and 2.73°) reflections were not indexed with these parameters. Taking into account one weak (44 times weaker than the strongest reflection) and diffuse (width is 1.56°) reflection, the $a'$ unit cell parameter doubles: $a''$ = $2a'$, $b''$ = $b'$, $c''$ = $c'$. However, the second reflection remained non-indexed with these parameters. A superstructure with the unit cell parameters $a'$, $b'$, $c'$ was also found for the solid solution with the composition $(Ce_{0.80}Gd_{0.20})Sc_3(BO_3)_4$ [84].

### 3.2.18. $LaSc_3(BO_3)_4$-«$ErSc_3(BO_3)_4$»-«$YbSc_3(BO_3)_4$» System

Durmanov et al. [35] selected the optimal compositions of the melt to obtain crystals by the Czochralski method (Table 2). According to the XRD analysis, all grown crystals have a monoclinic symmetry, probably the space group $C2/c$. Based on the previous studies [76], Durmanov et al. [35] suggested that the $Er^{3+}$ and $Yb^{3+}$ ions replace the $Sc^{3+}$ ones, and their maximum concentration in the initial melt should not exceed 12 at %, since a higher content may affect the cracking of crystals and change the symmetry (similar to solid solutions in the $LaSc_3(BO_3)_4$ – «$ErSc_3(BO_3)_4$» system).

Thus, huntite-family solid solutions of rare-earth scandium borates crystallize either in a monoclinic symmetry with the centrosymmetric space group $C2/c$ (in many works, the centrosymmetry is not confirmed) or in a trigonal one with the non-centrosymmetric space groups $R32$ (requires confirmation) or $P321$. In most cases, due to methodological difficulties of the XRD experiment, the site occupancies were not refined for the structures under investigation, especially when several cations with close atomic scattering factors occupy one crystallographic site. However, an elemental analysis and a crystallochemical approach made it possible to suggest the most likely composition from different compositions of solid solutions.

## 4. Crystallochemical Features of the Huntite Family

The variety of compositions and different symmetries found for the huntite-family compounds and solid solutions allows to make their classification and systematization based on crystallochemical phenomena such as isomorphism (an ability of a system to form equivalents), morphotropy (a change in a crystal structure for a regular series of compounds having similar formula composition), polymorphism (an adaptability of a structure to external influences), and the adjacent polytypism (an ability of a substance to crystallize in various modifications with different packing of structural elements along one axis). Surely, there are no clear boundaries between them, but in the first approximation, this approach allows identifying general structural features and finding fundamental differences in the group of compounds or solid solutions.

In addition to the possibility of forming continuous or limited solid solutions (the unlimited (perfect) and limited (imperfect) isomorphism) and internal ones (preferably for rare-earth scandium borates), a morphotropy is characteristic of the huntite-family compounds. The main reason for the morphotropy, in this case, is the size factor, namely, a change in the ionic radius of the $Ln$ cation ($LnAl_3(BO_3)_4$–$LnGa_3(BO_3)_4$–$LnSc_3(BO_3)_4$) (Figure 1) or $M$ cation ($LaAl_3(BO_3)_4$–$LaFe_3(BO_3)_4$–$LaCr_3(BO_3)_4$–$LaSc_3(BO_3)_4$, $TbAl_3(BO_3)_4$–$TbFe_3(BO_3)_4$–$TbCr_3(BO_3)_4$–$TbGa_3(BO_3)_4$–$TbSc_3(BO_3)_4$, $YbAl_3(BO_3)_4$–$YbFe_3(BO_3)_4$–$YbCr_3(BO_3)_4$–$YbGa_3(BO_3)_4$) (Figure 11) [114]). *The morphotropic series* include nominally-pure and activated samples, for example, $LaSc_3(BO_3)_4$ (space group $C2/c$) – $LaSc_3(BO_3)_4$:$Cr^{3+}$ (space group $P1$ or $P\bar{1}$) (an electronic structure factor) [76] and $LaSc_3(BO_3)_4$ (space group $C2/c$) – $LaSc_3(BO_3)_4$:$Nd^{3+}$ (space group $C2$) (a size factor) [24].

*The polymorphic transition* from the space group $P3_121$ to the $R32$ one (from a low-symmetric to a high-symmetric modification) with increasing temperature is known for the $Ln$Fe$_3$(BO$_3$)$_4$ with the $Ln$ = Eu–Er, Y [53,56–58,60,62–68] (Figures 1 and 13).

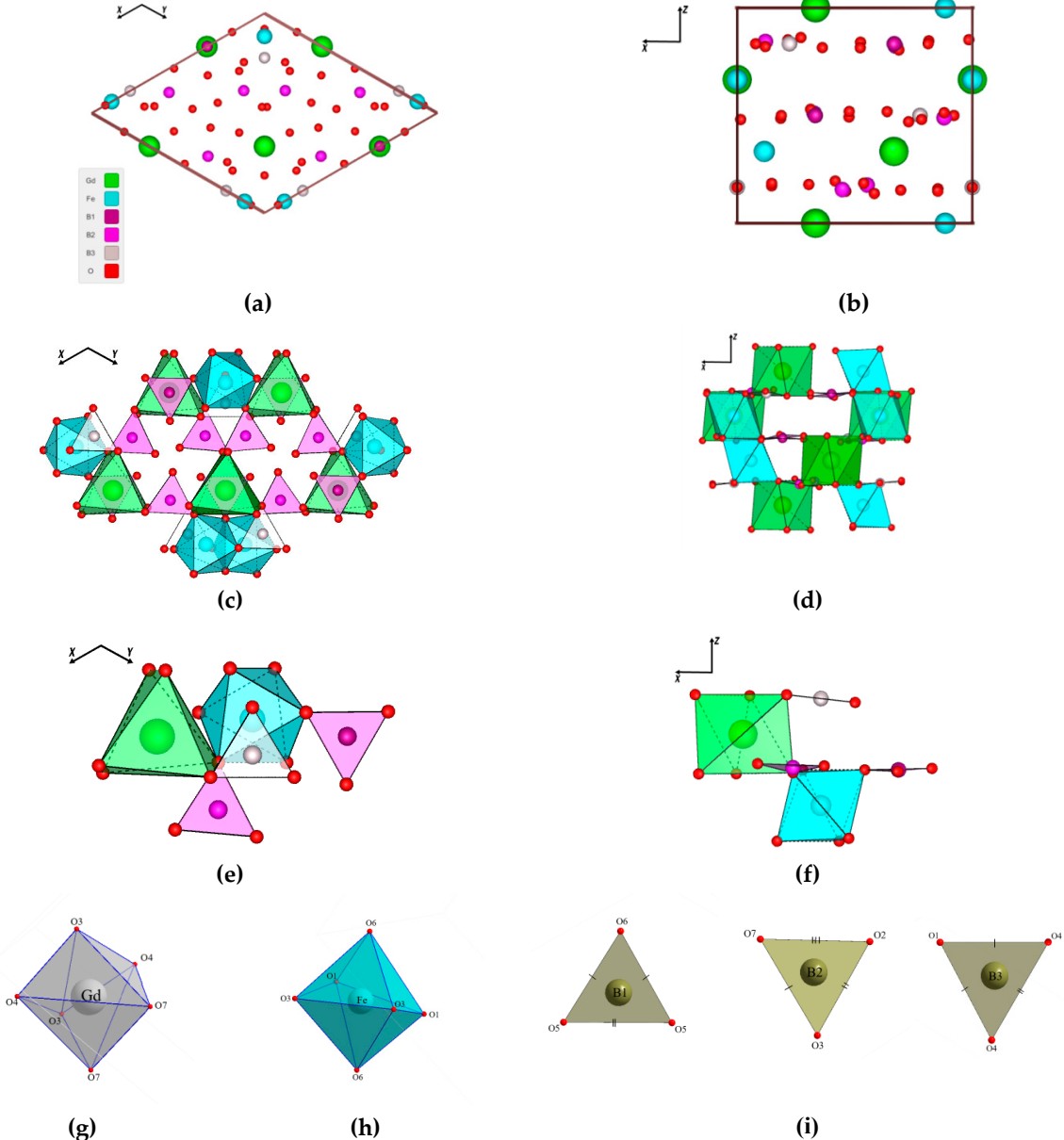

**Figure 13.** The unit cell of the GdFe$_3$(BO$_3$)$_4$ structure (space group $P3_121$) projected onto the (**a**) XY and (**b**) XZ planes; Combination of the coordination polyhedra projected onto the (**c**) XY and (**d**) XZ planes; Combination of selected coordination polyhedra projected onto the (**e**) XY and (**f**) XZ planes; Coordination polyhedra for the (**g**) Gd, (**h**) Fe, (**i**) B1–B3.

In the $Ln$Fe$_3$(BO$_3$)$_4$ structure with the low-temperature modification (space group $P3_121$) with the unit cell parameters (Figure 13a,b) similar to those for the huntite (space group $R32$) ($a = 9.5305$, $c = 7.5479$ Å for the GdFe$_3$(BO$_3$)$_4$), the polyhedra (Figure 13c–i) are the same as those in the structures of huntite-like compounds: a trigonal prism for the $Ln$ (Figure 13g), an octahedron for the Fe (Figure 13h), and three triangles for the B atoms (in the huntite structure, there are two crystallochemically-different B atoms) (Figure 13i). A comparison of the XZ projections of the structures with the space groups $R32$ (Figure 3b) and $P3_121$ (Figure 13b) shows a similar alternation of atoms along the Z axis (Ln, Sc–B,

O–Ln, Sc–B, O) and polyhedra (Figure 3d,f and Figure 13d,f), but with corresponding gaps in the structure with the space group $P3_121$ (compare Figures 3c and 13c and Figures 3d and 13d) due to the absence of the second helical axis.

The space group $P3_121$ was derived for the $GdFe_3(BO_3)_4$ structure (90 K) from the systematic extinctions and was discriminated from other candidate space groups that comply with the same extinction conditions during the structure determination process; the polarity of the structure actually chosen (space group $P3_121$) was determined by Flack's x refinement [63]. Using circularly polarized X rays at the Dy $L3$ and Fe $K$ absorption edges, Nakajima et al. [115] established that the single-crystal $DyFe_3(BO_3)_4$, which has the chiral helix structure of Dy and Fe ions on the screw axes, belongs to the left-handed space group $P3_221$, and this is in accord with the results on soft x-ray diffraction at Dy $M5$ absorption edge [116].

With an appropriate choice of the origin of the coordinates for the $LnFe_3(BO_3)_4$ unit cell (on the Gd atom, Figure 13b), the topological similarity of the structures with the space groups $R32$ and $P3_121$ is observed. At the same time, the high-temperature modification is more symmetrical than the low-temperature one, which is typical of polymorphism [117].

For phases with the compositions $(Ln,Sc)Sc_3(BO_3)_4$ and $LnSc_3(BO_3)_4$, a polymorphic "order–disorder" phase transition (if the space group $R32$ is proven for these compounds) from the space group $P321$ (a disordered structure with a partial or full ordering of the $Ln^{3+}$ and $Sc^{3+}$ ions over two trigonal-prismatic sites and that of the $Sc^{3+}$ ions over two octahedral sites) to the space group $R32$ (an ordered structure with a statistical arrangement of the $Ln^{3+}$ and/or $Sc^{3+}$ ions in one trigonal-prismatic site and that of the $Sc^{3+}$ ions in one octahedral site) can be assumed with increasing temperature.

On the other hand, it is possible that a kinetic 'order–disorder' phase transition (see, for example, [118,119]) occurs for rare-earth scandium orthoborates, i.e., a partially ordered phase (space group $P321$) is formed in the stability region of the disordered phase (space group $R32$) (a positional 'ordering–disordering') under an influence of kinetic (growth) factors. Growth dissymmetrization, as a rule, affects local parts of a crystal (hence, no more than 67 % of diffraction reflections, which did not correspond to the space group $R32$, were found), i.e., there is a different correlation between unit cells with different symmetries (a peculiar volume defect).

A necessary condition for this kind of ordering in the structure is, first of all, the presence of sites jointly occupied by several crystallochemically-different atoms and their concentration. For example, an increase in the content of Yb activator in the scheelite-family $(Na_{0.5}Gd_{0.5})MoO_4$:Yb crystals grown by the Czochralski method leads to an increase in the degree of structure deviation from the space group $I4_1/a$ and an increase in the orthorhombic distortion of the resulting superstructure [120]. For the $(Na_{0.5}Gd_{0.5})MoO_4$: 10% Yb crystals, ~50% of reflections, which are not inherent in the scheelite centrosymmetric space group $I4_1/a$, but typical for the non-centrosymmetric space group $P\bar{4}$, was revealed by the XRD experiment [120]. It should be noted that only ~ 4% of such reflections was found for the $(Na_{0.5}Gd_{0.5})WO_4$: 10% Tm crystals ($r_{Gd} > r_{Tm} > r_{Yb}$), and $(Na_{0.5}Bi_{0.5})MoO_4$ crystallizes in the space group $I\bar{4}$ [121]. An ordering depends on the growth method and synthesis conditions (for the Czochralski method: crystallization, cooling, and annealing rates; annealing and quenching temperatures; growth atmosphere, etc.). Indeed, an increase in growth rates from 4 to 6 mm/hour reduces a degree of structure ordering for the $(Na_{0.5}Gd_{0.5})WO_4$ crystal. The cooling rate of the crystals has a similar effect: a decrease in the cooling rate contributes to the formation of ordered non-centrosymmetric $(Na_{0.5}Gd_{0.5})WO_4$ and $(Na_{0.5}Gd_{0.5})WO_4$:Yb crystals (space group $I\bar{4}$) [122]) Hence, a growth disymmetrization is most pronounced in conditions close to equilibrium.

According to [46,123], for the $LnAl_3(BO_3)_4$ compounds, a modification with the space group $R32$ is formed at low temperatures (~880–900 °C), phases with the centrosymmetric $C2/c$ symmetry crystallize in the higher temperature region, up to 1040–1050 °C, and a further temperature increase leads to a formation of the most disordered and metastable non-centrosymmetric modification with the $C2$ symmetry. It can be seen that structurally disordered modifications with low symmetry are

formed at elevated temperatures, which contradicts the rules of *polymorphism*, but it is characteristic of polytypism [124]. Inclusions of one polytype in another were revealed for the $LnAl_3(BO_3)_4$ with the $Ln$ = Nd and Gd by the IR spectroscopy using a factor group analysis for vibrations of the B–O bond [45]. Moreover, the structure of the monoclinic $SmAl_3(BO_3)_4$ crystal contains significant fragments of the trigonal polytype, and the structure of the trigonal $NdAl_3(BO_3)_4$ have a high content of domains of the monoclinic polytype. However, this conclusion was not confirmed by structural studies.

The unit cell parameters of the huntite structure with the space group $R32$ (the hexagonal cell: $a_{R32} = b_{R32} \neq c_{R32}$) and those of the huntite-family structures are related as (Figure 14):

- $a_{C2/c} = 0.666[c^2_{R32} + (a_{R32}\cos30°)^2]^{1/2}$, $b_{C2/c} = b_{R32}$, $c_{C2/c} = 0.666 [(2c_{R32})^2 + (a_{R32}\cos30°)^2]^{1/2}$ (space group $C2/c$: $a = 7.7297(3)$, $b = 9.8556(3)$, $c = 12.0532(5)$ Å, $\beta = 105.405(3)°$) [82];

- $a_{C2/c} = 0.666[c^2_{R32} + (a_{R32}\cos30°)^2]^{1/2}$, $b_{C2/c} = b_{R32}$, $c_{C2/c} = [(2.5c_{R32})^2 + (a_{R32}\cos30°)^2]^{1/2}$ (space group $C2/c$: $a = 7.227(3)$, $b = 9.315(3)$, $c = 21.688(3)$ Å, $\beta = 95.90(2)°$) [97];

- $ac_2 = 0.666[c^2_{R32} + (a_{R32}\cos30°)^2]^{1/2}$, $bc_2 = b_{R32}$, $c_2 = 1.333[(1.25c_{R32})^2 + (a_{R32}\cos30°)^2]^{1/2}$ (space group $C2$: $a = 7.262(3)$, $b = 9.315(3)$, $c = 16.184(8)$ Å, $\beta = 90.37°$) [47].

It follows that two unit cell parameters are the same for the huntite-like structures, and the third $c$ parameter is correlated with the other two.

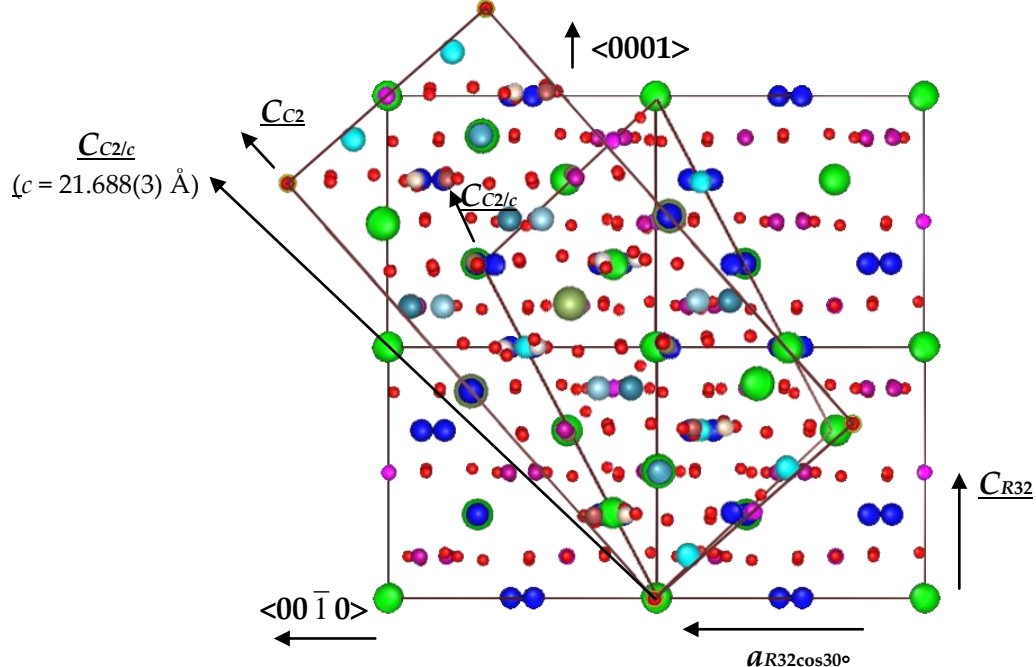

**Figure 14.** Genetic correlations between the unit cells for the huntite family compounds.

According to the results of the XRD analysis of the micropart of the crystal with the initial composition $Nd_{1.25}Sc_{2.75}(BO_3)_4$ (NSB–1.25), the unit cell parameters determined by auto-indexing of 21 reflections ($h0\bar{h}0$, $000l$) in the interval of interplanar distances $d = 2.01$–8.14 Å, correspond to the primitive trigonal cell with the $c_{R32}$ parameter doubled with respect to the huntite one ($a = a_{R32} = 9.74$, $c = 2c_{R32} = 15.83$ Å) [89]. In the interval of interplanar distances $d = 3.96$–4.00 Å, several diffuse reflections with a width of 1.23–1.4° were found (the remaining reflections of approximately the same intensity had a width of 1.05°). Taking them into account (25 reflections), a primitive trigonal cell with the doubled $a_{R32}$ and $c_{R32}$ parameters ($A = 2a_{R32} = 19.526(3)$, $C = 2c_{R32} = 15.838(2)$ Å), as for the $(Ce_{0.41(4)}Nd_{0.46}Gd_{0.13})Sc_3(BO_3)_4$, was obtained. It should be noted that the refinement of the NSB–1.25 composition in the space group $R32$ by the Rietveld method allowed to find its composition as $NdSc_3(BO_3)_4$. An analysis of the atom displacement and positional parameters (first of all, for

the B2 atoms) indicates a structure different from the huntite with the space group *R*32. A similar diffraction picture was observed for the solid solutions in $CeSc_3(BO_3)_4$ - $NdSc_3(BO_3)_4$ - «$GdSc_3(BO_3)_4$» and $CeSc_3(BO_3)_4$ - $NdSc_3(BO_3)_4$ - «$LuSc_3(BO_3)_4$» systems.

As a result of quenching of rare-earth aluminum orthoborates with the trigonal symmetry, in addition to the huntite-type modification (space group *R*32), a structural state with a full structural disorder in the alternation of layers along the *c* axis was revealed in the single-crystal diffraction patterns (diffuse bands on the reciprocal lattice along the *c** parameter) [46]. In the diffraction patterns of monoclinic phases (space groups *C*2/*c* and *C*2), weak diffuse reflections, which should be absent due to extinction of space group reflections of the matrix structures, were also observed.

All the above-mentioned experimental facts, namely, a presence of diffuse areas along with the point ones in the diffraction patterns, a presence of strong diffraction reflections with a high symmetry, the similar $a_{R32}$ and $b_{R32}$ parameters and the $c_{R32}$ parameter, which can be represented as a linear combination of vectors (for classical polytypes, the similar *a* and *b* parameters and the *c* one multiple to minimum) meet the general principles of polytypism and are optimally described in terms of OD (order–disorder) theory [125].

The main factors for the formation of polytypes for the $LnAl_3(BO_3)_4$ are a crystallization temperature (thermodynamic factor), a crystallization rate and a cooling rate in the flux method (kinetic factor), and a nature of the *Ln* cation and a $r_{Ln}/r_{Al}$ ratio (crystallochemical factor) [27]. For the $LnAl_3(BO_3)_4$ with the *Ln* = Pr – Gd, two polytypic modifications with the space groups *R*32 and *C*2/*c* occur: for the *Ln* = Pr and Nd, a monoclinic structure is more stable; for the *Ln* = Sm, Eu, and Gd, a trigonal modification is more stable, a monoclinic phase is formed in the flux at high temperatures and concentrations only. In the case of orthoborates with the *Ln* = Tb - Lu and Y, a modification with the huntite structure is only stable [123]. Beregi et al. [97] concluded that the starting crystallization temperature is a dominant factor in formation of the trigonal and monoclinic symmetry: for the phase with the nominal composition $Eu_{0.02}Tb_{0.12}Gd_{0.86}Al_3(BO_3)_4$, a higher (1080 °C) and lower (1060 °C) starting temperatures led to an appearance of modifications with the space groups *C*2/*c* (*a* = 7.227(3), *b* = 9.315(3), *c* = 21.688(3) Å, β = 95.90(2)°) and *R*32 (*a* = 9.294(2), *c* = 7.251(2) Å), respectively.

Rare-earth scandium orthoborates, unlike other huntite-family rare-earth borates, exhibit a slightly different structural behavior. For example, for the $LaSc_3(BO_3)_4$, three modifications are known: a low-temperature (space group *Cc*, is a subgroup of the space group *C*2/*c* and it is absent for other rare-earth orthoborates), a medium-temperature (space group *R*32; as was mentioned above, this group is denied by many researchers) and a high-temperature (space group *C*2/*c*) ones. The order of realization of the symmetry by the scandium borate crystals with increasing temperature is clearly different from that found for rare-earth aluminum borates: the most symmetrical structure crystallizes at high temperatures. It is quite possible that rare-earth scandium orthoborates are characterized primarily by polymorphs, although polytypes are also possible.

## 5. Conclusions

A critical analysis of the data on growth and structural diagnostics (composition, structure) of the huntite-family compounds and solid solutions indicates the most complete and consistent data obtained for rare-earth aluminum orthoborates. Based on the results of investigation of the $LnAl_3(BO_3)_4$ crystals with the *Ln* = Pr – Lu, Y obtained by the flux method (spontaneous crystallization and crystallization on a seed) with different solvents, a theory of polytypism is expanded (for example, [46]) due to the fact that all modifications known to date have been obtained and structurally characterized for aluminum orthoborates. The largest number of publications is devoted to activated and co-activated $YAl_3(BO_3)_4$ crystals with the space group *R*32. Monoclinic modifications are referred less often, and their centrosymmetry or non-centrosymmetry was not proved but only stated for almost all rare-earth orthoborates. In addition, structural single-crystal studies for the $LnCr_3(BO_3)_4$ having also a variety of modifications are absent to date, a symmetry being affected by the borate:solvent ratios in the batch during the spontaneous crystallization from a flux (Figure 1).

Crystal structures for almost all phases were determined (refined) using the X-ray experiment (there are several works on the neutron experiment, in particular [56,61,66–68]). The occupancies for the *Ln* and *M* sites were not refined (with a few exceptions), they often were fixed to those determined from the ICP elemental analysis. The real composition, obtained by refining the *Ln* and Sc site occupancies in structures of the $LnSc_3(BO_3)_4$ crystals with trigonal and monoclinic symmetries, is generally not coincide and coincide with the charge composition, respectively. As a result, a congruent melting for the monoclinic phases is possible.

Many questions remain to the rare-earth scandium orthoborates, since it is possible to find literature data with the opposite results on existence or absence of different modifications, in particular, a realization of the trigonal phases with the space group *R*32 or *P*321. This is primarily due to the fact that these phases are obtained by different methods under different conditions, and quite possibly, this modification can be obtained by a specific growth method under specific synthesis conditions. We failed to obtain the huntite-family $LnSc_3(BO_3)_4$ compounds with the *Ln* = Sm and Gd by the solid-phase synthesis of the corresponding oxides at the $T$ = 1000, 1250, 1500 °C and $T$ = 1500 °C, respectively, although in the literature, there is data on the synthesis of the $LnSc_3(BO_3)_4$ compounds with the *Ln* = Sm, Eu, Gd, Y. An X-ray study of single crystals with the nominal composition $LnSc_3(BO_3)_4$ with the *Ln* = La, Ce, Pr, Nd, Tb and numerous solid solutions, grown by the Czochralski method, with the subsequent crystallochemical analysis of the results obtained, makes it possible to doubt the possibility of obtaining the $LnSc_3(BO_3)_4$ with *Ln* = Sm, Gd, Eu, Y by this method. Although internal solid solutions with these components are possible to obtain by a careful selection of the initial charge compositions and synthesis conditions.

When crystals are grown by the Czochralski method from melts at high temperatures, their symmetry depends on a composition and sintering temperature of a charge; a type, composition, symmetry, and orientation of a seed; a growth atmosphere, rotation, and pulling rates, as well as on an efficiently control both processes of $B_2O_3$ vapors condensation on the growing crystal and temperature gradients above the crucible and in the melt [35]. It should be noted that the $LnSc_3(BO_3)_4$ can easy change its symmetry (group–subgroup), therefore a structural experiment should be performed on single-crystal objects with the analysis of diffraction reflections, including low-intensity ones (the use of high-resolution transmission electron microscopy and synchrotron radiation is also can be carried out), and the refinement of crystallographic site occupancies, i.e., real crystal composition. The refinement of structures by the full-profile method is correct only when using a developed methodology with criteria formulated to attribute the structure to the space group *R*32 or *P*321. This is important for establishing correlations between symmetry, real composition of objects and growth methods and conditions; correct explanation of functional properties observed; direct growth of crystals with a required combination of physical parameters, as well as for clarifying and summarizing crystallochemical data for the huntite family compounds and other functional materials.

## 6. Patents

Kuz'micheva, G.M.; Podbel'sky, V.V.; Chuykin, N.K.; Kaurova, I.A. Program for the Investigation of the Dynamics of Changes in the Structural Parameters of Compounds with Different Symmetry; Certificate of state registration of computer software no. 2017619941: Moscow, Russia, 12 September 2017 (in Russian).

**Author Contributions:** Conceptualization, G.M.K.; Methodology, G.M.K. and V.B.R.; Software, V.V.P.; Validation, G.M.K., I.A.K. and V.B.R.; Formal Analysis, G.M.K., I.A.K. and V.B.R.; Investigation, G.M.K. and V.B.R.; Resources, G.M.K., I.A.K., V.B.R. and V.V.P.; Data Curation, G.M.K. and I.A.K.; Writing—Original Draft Preparation, G.M.K. and I.A.K.; Writing—Review & Editing, G.M.K.; Visualization, V.V.P.; Supervision, G.M.K. and I.A.K.; Project Administration, I.A.K.

**Funding:** This research received no external funding.

**Conflicts of Interest:** The authors declare no conflict of interest.

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
