# Peer review of "Crystallochemical Design of Huntite-Family Compounds"

_crystals, doi:10.3390/cryst9020100_

Round 1
Reviewer 1 Report
The review devoted to summarizing everything known at present about huntite-family nominally-pure and activated/co-activated LnM3(BO3)4 (Ln = La-Lu, Y; M =Al, Fe, Cr, Ga, Sc) compounds and their-based solid solutions.
This review should be considerably revised before publication. The text of the review can be reduced as follows:
1. Table 2-7 contain already published data (coordinates of atoms, thermal parameters and interatomic distances) which are not discussed in the text at all. The reader can find these data both in the corresponding references ([41, 79, 82, 88, 97]), and in the Inorganic Crystal Structure Database (ICSD)
2. The majority of structure projections and coordination polyhedrons presented in fig. 3-9 and fig. 13 are also not discussed in the text and are unnecessary. It would therefore be wiser to edit these figures according to contents of the review
3. The text of Section 3.2 contains a lot of the repeating expressions and terms. There is a wish to recommend to authors to summarize data in the table with columns: compositions, unit cell parameters, space group, synthesis method, reference

Author Response
Dear Reviewers,
We are thankful and express gratitude for reviewing the manuscript “Crystallochemical Design of Huntite-family Compounds”, by Galina M. Kuz’micheva, Irina A. Kaurova, Victor B. Rybakov, and Vadim V. Podbel’sky.
We are confident that your comments will help to make the manuscript more significant.
We have revised and checked the manuscript.
Please find below the detailed explanation of changes made.
1. Table 2-7 contain already published data (coordinates of atoms, thermal parameters and interatomic distances) which are not discussed in the text at all. The reader can find these data both in the corresponding references ([41, 79, 82, 88, 97]), and in the Inorganic Crystal Structure Database
Comments:
Tables 2-7 have been deleted.
New Tables 3 and 4, containing non-published structural data for the (Ce,Gd)Sc3(BO3)4 solid solutions, have been added.
2. The majority of structure projections and coordination polyhedrons presented in fig. 3-9 and fig. 13 are also not discussed in the text and are unnecessary. It would therefore be wiser to edit these figures according to contents of the review
Comments:
Additional text, described the Figures 3-9 & 13, has been added.
In the revised manuscript, the additional information is marked with the yellow. Please see Lines:
318-322, 404-407, 435-443, 451-453, 488-491, 493-496, 511-520, 541-543, 550-556, 883-887.
3. The text of Section 3.2 contains a lot of the repeating expressions and terms. There is a wish to recommend to authors to summarize data in the table with columns: compositions, unit cell parameters, space group, synthesis method, reference
Comments:
Table 2 with summarized structural data for rare-earth scandium borate solid solutions have been added.
The text of Section 3.2 has been reduced.
Reviewer 2 Report
The manuscript present a comprehensive review on the growth conditions and structural properties of borates with various compositions. This work will be useful for scientists working with these materials. Therefore, I would like to recommend the publication of the manuscript after the authors consider the comments below:
- From the text it is not quite clear what is the analysis of literature data and what is the own contribution. Examples: sentence in lines 19-22 (abstract), using of the structures like “were refined” (e.g. lines 219, 241, 258).
- The majority of information presented (6 tables out of 7) is devoted to Sc-based borates. This fact should be more clearly reflected in the abstract.
- Technical info in lines 279-280 is unnecessary.
- Figs. 10, 12 are too small.
- Caption to Fig. 12: the abbreviations should be specified.
Author Response
Dear Reviewers,
We are thankful and express gratitude for reviewing the manuscript “Crystallochemical Design of Huntite-family Compounds”, by Galina M. Kuz’micheva, Irina A. Kaurova, Victor B. Rybakov, and Vadim V. Podbel’sky.
We are confident that your comments will help to make the manuscript more significant.
We have revised and checked the manuscript.
Please find below the detailed explanation of changes made.
1. From the text it is not quite clear what is the analysis of literature data and what is the own contribution. Examples: sentence in lines 19-22 (abstract), using of the structures like “were refined” (e.g. lines 219, 241, 258).
Comments:
In the present manuscript, an analysis of literature data, including the data obtained by our scientific group earlier, and new results are shown.
To distinguish results obtained by our scientific group, in the manuscript, we have noticed “present work” (see, for example, Table 2) or gave references to our previous works (see, for example, lines 260-261).
In addition, Tables 3 & 4 with new structural data obtained by our scientific group have been added.
2. The majority of information presented (6 tables out of 7) is devoted to Sc-based borates. This fact should be more clearly reflected in the abstract.
Comments:
An additional sentence has been added to the Abstract (see Lines 22-24).
3. Technical info in lines 279-280 is unnecessary.
Comments: Technical info in lines 279-280 has been deleted.
4. Figs. 10, 12 are too small.
Comments:
Figs. 10 & 12 have been improved.
5. Caption to Fig. 12: the abbreviations should be specified.
Comments:
In Fig. 12, the abbreviations have been specified.
Round 2
Reviewer 1 Report
The authors made the revising and addition in the manuscript. Now it would be desirable to recommend to authors to remove an inaccuracies and misprint in the text.
1. It is necessary to examine all formulas on p. 13 in second and third paragraphs and p.15 in first paragraph (for example: Ca2+Mg2+3(CO3)4+4 and Ln3+M3+3(BO3)3+4 etc)
2. Instead of “the Ca-Mg-CO3 system” ( p.15 and 29) and La-Sc-(BO3) (p.29) it would be more correct to write CaCO3-MgCO3 system and La2O3-Sc2O3-B2O3 system
3. It is necessary check the axis X labelling on fig. 11 (rLnVI – rMVI, ?)
4. It is necessary to examine all formulas for the scheelite-family crystals on p.49, fourth paragraph: instead (Na0.5Gd0.5)MoO3:Yb there have to be (Na0.5 Gd0.5)MoO4:Yb etc.
5. In fig. 14 it is desirable to give different designations for two various the monoclinic space group C2/c corresponding to the text on p. 50
6. For reduction of volume of table 2 it would be desirable to take out the repeating text “a structure and composition are refined in the space group R32” in the note to table 2

Author Response
Dear Reviewer,
We are thankful and express gratitude for additional reviewing the manuscript.
We have revised and checked the manuscript.
Please find below the detailed explanation of changes made.
Point 1: It is necessary to examine all formulas on p. 13 in second and third paragraphs and p.15 in first paragraph (for example: Ca2+Mg2+3(CO3)4+4 and Ln3+M3+3(BO3)3+4 etc)
Response 1: Formulas have been improved.
Point 2: Instead of “the Ca-Mg-CO3 system” ( p.15 and 29) and La-Sc-(BO3) (p.29) it would be more correct to write CaCO3-MgCO3 system and La2O3-Sc2O3-B2O3 system
Response 2: Formulas have been improved.
Point 3: It is necessary check the axis X labeling in fig. 11 (rLnVI – rMVI, ?)
Response 3: The axis X labeling in fig. 11 has been improved, D(rLnVI – rMVI), Å.
Point 4: It is necessary to examine all formulas for the scheelite-family crystals on p.49, fourth paragraph: instead (Na0.5Gd0.5)MoO3:Yb there have to be (Na0.5 Gd0.5)MoO4:Yb etc.
Response 4: Formulas have been improved.
Point 5: In fig. 14, it is desirable to give different designations for two various the monoclinic space group C2/c corresponding to the text on p. 50
Response 5: In Fig. 14, an additional comment (the c unit cell parameter) has been given to distinguish two various monoclinic space groups C2/c.
Point 6: For reduction of volume of table 2 it would be desirable to take out the repeating text “a structure and composition are refined in the space group R32” in the note to table 2
Response 6: Note 3 has been given: A structure and composition were refined in the space group specified